**Subject Category:**
Biology (whole organism)

ecology/plant science/health and disease and epidemiology

dry matter content, fruit quality, *Malus domestica*, minerals, pollination, storage time

**Author for correspondence:**
Ulrika Samnegård
e-mail: ulrika.samnegard@biol.lu.se; ulrika.samnegard@gmail.com

†Present address: School of Environmental and Rural Sciences, University of New England, Armidale, New South Wales, Australia.

# Pollination treatment affects fruit set and modifies marketable and storable fruit quality of commercial apples

Ulrika Samnegård[1,2,†], Peter A. Hambäck[3] and Henrik G. Smith[1,2]

[1]Centre for Environmental and Climate Research, and [2]Department of Biology, Lund University, 223 62 Lund, Sweden
[3]Department of Ecology, Environment and Plant Sciences, Stockholm University, 106 91 Stockholm, Sweden

US, 0000-0002-3791-4688; PAH, 0000-0001-6362-6199; HGS, 0000-0002-2289-889X

Insect-mediated pollination increases yields of many crop species and some evidence suggests that it also influences crop quality. However, the mechanistic linkages between insect-mediated pollination and crop quality are poorly known. In this study, we explored how different pollination treatments affected fruit set, dry matter content (DMC), mineral content and storability of apples. Apple flowers supplementary pollinated with compatible pollen resulted in higher initial fruit set rates, higher fruit DMC and a tendency for lower fruit potassium (K) : calcium (Ca) ratio than flowers that received natural or no pollination. These variables are related to desirable quality aspects, because higher DMC is connected to higher consumer preference and lower K : Ca ratio is related to lower incidence of postharvest disorders during storage. Using structural equation modelling, we showed an indirect effect of pollination treatment on storability, however mediated by complex interactions between fruit set, fruit weight and K : Ca ratio. The concentrations of several elements in apples (K, zinc, magnesium) were affected by the interaction between pollination treatment and apple weight, indicating that pollination affects element allocation into fruits. In conclusion, our study shows that pollination and the availability of compatible pollen needs to be considered in the management of orchard systems, not only to increase fruit set, but also to increase the quality and potentially the storability of apples.

# 1. Introduction

Pollination services increase fruit and seed set of many crops, in particular that of vegetables, fruits, berries and nuts [1]. For this reason, it is well recognized that animal-mediated pollination is important for global food production in general and human nutrition in particular [1–3]. However, accumulating evidence shows that the positive effect of pollination services goes beyond increasing fruit and seed set, and may also increase quality (rapeseed, strawberries; [4,5]), affect nutritional composition (almonds, apples; [6,7]), decrease malformations (Fuji apples, strawberries; [8,9]) and increase shelf lifetime (strawberries; [4,9]) of crops. However, the linkages between pollination services and fruit quality have only just started to be recognized and there is a need for research that simultaneously evaluate several quality aspects and their interrelated effects.

Apple is a fruit crop with a strong dependence on animal-mediated pollination, in which all cultivars are self-incompatible to some extent and therefore require pollen transfer from another pollinizer cultivar to set commercially acceptable fruit levels [10]. It is the most geographically widespread temperate fruit [11] and the most common pollinator-dependent crop in Europe, where economic gains from pollination-induced increases in fruit set are higher than those of any other crop [12]. However, for commercially produced fruits, not only the quantity of fruit matters, because marketable fresh fruits also need to be of adequate quality for good storability and to attract consumers. Even though higher insect pollinator levels are known to increase pollinator activity and cross-pollination between cultivars and thus improve fruit yields of apples [13,14], the influence of insect pollination on quality aspects are more equivocal [7].

Several quality aspects have large economic impact on apple production. Important quality attributes for consumers include flavour and flesh firmness. Firmer fruits are considered by consumers to have higher quality [15,16]. Flavour is a complex attribute that is related to the dry matter content (DMC) in fruits, where higher DMC generally increases consumer preference [16,17]. The DMC also influences the firmness of apples at harvest and softening rates during storage, and is a good estimate of total soluble solids after storage [16,18,19]. Another important quality aspect for producers and wholesalers is storability. Apples can be stored for protracted periods of time, which allows for longer market availability. However, even though the storage facilities have developed substantially, which has prolonged storability, much fruit is still discarded when taken out from storage. For example, in Sweden, 20% of organic apples were disregarded in an experimental study in 2010 and, depending on cultivar, 9–27% of conventionally produced apples from seven orchards were disregarded after storage during 2010–2015 (I. Tahir 2017, personal communication). Many aspects influence the storability of apples, where harvest time and mineral concentrations are important modifiers. For example, low calcium (Ca) content, and the high ratio between magnesium (Mg) or potassium (K) and Ca, is connected to postharvest disorders including bitterpit, lenticel breakdown and Jonathan spot [20–24]. Ca and its ratio with other elements (e.g. K:Ca and nitrogen (N):Ca) is also connected to the softening of apples and resistance to diseases [25–27]. Consequently, Ca application both before and after harvest to increase fruit Ca-content is a common management action in modern orchards [22,26].

A few studies have suggested that apples which successfully have been cross-pollinated differ in their mineral content from less pollinated ones, suggesting that the mechanisms influencing mineral allocation into fruits are related to pollination services. Porcel *et al.* [28] found a positive relationship between seed number and Ca, K and Mg content in Aroma apples, Bramlage *et al.* [29] found a positive relationship between seed number and Ca concentrations in Richared Delicious apples and Volz *et al.* [30] found that supplementary pollination positively affected Ca concentrations of Braeburn apples. Other data suggest that the effect of pollination and fertilization on Ca concentrations may be cultivar-specific. Buccheri & Di Vaio [31] found a positive relationship between higher seed set and Ca concentration in fruits from some cultivars (Red and Golden Delicious) but not from other (Annurca Rossa del Sud and Annurca Tradizionale), and Garratt *et al.* [7] found that supplementary hand-pollination even decreased Ca concentrations in Gala apples. Because the mineral content may be related to other quality aspects including firmness, postharvest disorders and storability, pollination service may have a more far-reaching role in the economy of apple production than has earlier been estimated.

The aim of this study was to simultaneously evaluate the direct effects from different pollination treatments on the mineral concentration in apples, the marketable fruit quality of apples and the indirect effects on the storable fruit quality. The pollination treatments—supplementary hand-pollination, natural pollination and pollinator exclusions—were related to a decreasing probability of

successful cross-pollination, where the supplementary hand-pollinated flowers had the highest and the pollinator-excluded flowers the lowest probability of successful cross-pollination. As an estimate of marketable fruit quality, we used DMC as an endpoint variable (cf. [16]), and as estimates of storable fruit quality, we used both the probability of developing various storage disorders and the time that the fruits could be stored and still maintain good quality. Our working hypothesis was that pollination treatment affects element concentrations, DMC and storability of fruits, where fruits from the supplementary hand-pollinated followed by the natural pollination treatment will have higher values than the pollinator-excluded fruits. We used a structural equation model (SEM) to disentangle the direct and indirect effects of pollination treatment on storability, and particularly the K : Ca ratio which has previously been implicated as a measure of storable fruit quality.

# 2. Methods

## 2.1. Sites

Two orchards were selected in the apple growing area of Kivik, southern Sweden, which is the main apple growing region in Sweden. The orchards were separated by 8 km and were both organically certified (by Swedish KRAV), thus not using synthetic fertilizers or pesticides. The orchards were irrigated and honeybee hives (one colony ha$^{-1}$) and commercial bumblebees (0.5 colony ha$^{-1}$) were present during apple bloom. A mix of apple cultivars were planted in the orchards, both compatible cultivars for production and specific pollinizer varieties. The apple harvests from the orchards are sold in the market or are used for production of apple juice and cider.

## 2.2. Pollination treatment

Within both orchards, rows with the apple cultivar Amorosa, a red subcultivar of Aroma, were identified. Aroma/Amorosa are among the most popular apple cultivars in Swedish orchards and were the most commonly planted cultivars in new plantations in 2012 [32,33]. Distributed over two rows, 40 and 30 trees were marked in the two orchards, respectively. On each tree, three branches at similar height were marked and randomly assigned to one of the following treatments: pollinator exclusion, supplementary hand-pollination and natural pollination. When the flower buds were approaching balloon stage, the pollinator exclusion branches were bagged with 255 × 610 mm perforated Crispac-bags, plastic bags permeable to air through small holes (∅ = 0.5 mm), to exclude pollinators. All flowers on the branch were inside the bag. In the few cases when the bag did not cover all flower clusters, clusters outside the bag were removed. Bags were removed when all flowers had withered. At peak flowering in late May 2016, 5–10 flowers on the supplementary hand-pollination branch, in addition to natural pollination from the present pollinator community, received supplemented hand-pollination using a cotton swab with fresh pollen collected from at least three different trees of a compatible cultivar (Holsteiner–Cox). Flowers that had received supplementary pollen were marked. The flowers on the natural pollination branches received only visits from the present pollinator community. The number of flowers on the branches assigned to the different treatments as well as the total number of flowers on the whole tree were noted.

## 2.3. Fruit set

Approximately one month after peak flowering, the developing apples on the marked branches were counted as a measure of the initial fruit set. Some days before commercial harvest, 3.5 months after peak flowering, fruits were again counted on the marked branches. All developed fruits attached to the branches were counted, except for fully rotten fruit. The finest apples, without any major visible damage, were harvested from the branches. On the supplementary hand-pollinated branch, only marked fruits were harvested. To compensate for the lack of fruits on some branches, up to four apples were taken from branches of the same treatment with more fruits. In one orchard, fruit set was extremely low (presumably affected by the high infestation of the rosy apple aphid, *Dysaphis plantaginea*), and extra control apples were picked from unmarked trees, hereafter called 'extras'. In total, 244 apples were picked, 62 pollinator excluded, 55 supplementary hand-pollinated, 101 control and 26 extra apples.

## 2.4. Quality measurements

All collected apples were taken to the laboratory for measurements. All apples were weighed and their length, maximum and minimum diameter were measured using digital callipers. Ground colour was estimated using a chartreuse colour chart where '1' was darkish green and '8' was yellow coloration of the skin, per cent cover colour with red pigmentation was noted and if there were any pest damages or visible diseases on the apples. If these measurements were not taken on the same day as the harvesting date, the fruits were put in the refrigerator for a maximum of 3 days before measurements.

Following these initial measurements, apples were divided into two groups, where one ($n = 84$) was selected for additional destructive measurements and the other ($n = 160$) was used for measuring the storability and the quality of stored apples. The apples that were first stored were subjected to the same destructive measurements either when they were experiencing postharvest disorders or after 161–162 days when the experiment ended. The stored fruits were checked every second week and the fruits that could no longer be regarded as first-class fruit, because of postharvest disorders, were taken out from storage and measured. The fruit that started to shrivel was kept in storage if no other disorders were seen. During storage, the fruit was wrapped individually with paper, placed in perforated plastic backs commonly used for fruit packing and placed in a 6°C refrigerator.

The destructive measures taken on all apples included firmness using a penetrometer (Model FT-327; Effegi, Italy; plunger diameter 8 mm), per cent sucrose (°Brix/soluble solids content) using an eclipse handheld refractometer (Bellingham + Stanley Ltd) and counts of developed seeds. Following these measurements, fruits were frozen in a −20°C freezer for later analyses of acidity, DMC and mineral content. The mineral content analyses were carried out with an inductively coupled plasma-optical emission spectrometry (ICP-OES), Optima 8300, Perkin Elmer. Seven elements were analysed: K, phosphorus (P), Mg, Ca, Boron (B), Iron (Fe) and Zinc (Zn). Both dry and fresh weight concentrations were measured, but only fresh weight concentrations ($\mu g \, g^{-1}$ fresh weight) were used for statistical analyses. The concentration of the elements was adjusted according to the weight loss of each apple in storage. Two apples lacked a final weight measurement after storage, for those an estimated weight loss was calculated based on the mean weight loss per day for the other apples. Titratable acidity (TA) in the apples was measured by extracting 5 ml of apple juice, diluting the juice with 15 ml ddH$_2$O and titrating as malic acid with 0.05 N NaOH until a pH of 8.1 was reached. For the apples in which extraction of flowing juice was not possible, 5.0 g apple sauce was used instead and the acidity was corrected accordingly (5 ml juice = 5.25 g). Two fruits had started to rot and were disregarded before dry matter and mineral content analyses, and an additional three fruits were too small to analyse for both minerals and TA. Thus, 244 fruits were measured for initial measurements, 242 fruits for DMC and mineral content and 239 for TA.

## 2.5. Statistical analyses

Before any statistical analyses, all apples were checked to see if they meet the European Union marketing standards for apple (https://ec.europa.eu/agriculture/fruit-and-vegetables/marketing-standards_en, downloaded 9 January 2018). Twenty-six fruits were disregarded because they were too small (lighter than 70 g) or because their Brix value was too low (if lighter than 90 g, the Brix should be above 10.5).

Differences between pollination treatments for both initial and final fruit set were analysed by fitting generalized least-squares (GLS) models, using the nlme-package [34] in R [35]. Per cent initial and final fruit set were calculated by dividing the number of fruitlets and ripe fruits, respectively, with the number of initial flowers on each branch and multiplying it by 100. Per cent initial and final fruit set were included as response variables, and treatment and site as fixed factors. Final fruit set was square root transformed to meet the assumption of normally distributed model residuals. Models allowing for unequal variance between pollination treatments were used (*VarIdent* option) because they had better fit (initial fruit set Δ Akaike information criteria (AIC) = 116, final fruit set ΔAIC = 49). If a significant treatment effect was found using a likelihood-ratio test, we performed post hoc tests using the glht-function from the 'multcom package' in R [36], with Holm-adjusted *p*-values, with the predefined contrasts natural pollination—supplementary hand-pollination treatment and natural pollination—pollinator exclusion treatment. To make the glht-function to work with a GLS model, an extra function from http://rstudio-pubs-static.s3.amazonaws.com/13472_0daab9a778f24d3dbf38d808952455ce.html, downloaded 9 January 2018, was used. Because we had two measurements of fruit set from the supplementary hand-pollination treatment, we repeated the analyses to separately analyse the fruit set

from the subset of flowers that had received supplementary hand-pollination and the fruit set for the entire branch. The results from the entire branch are presented in the electronic supplementary material, figure S1.

Pollination treatment effects on seed set were analysed by fitting a generalized linear mixed-effects model, in the lme4 package [37], with the binomial response variable seed set (maximum 10 developed seeds per fruit), treatment as fixed factor and apple tree identity (ID) as random effect. An observation-level random effect was added to account for overdispersion [38]. If there was a significant treatment effect, we performed post hoc tests as above. Extra apples were not included in the analysis and one sample lacked seed count ($n = 191$).

The effects of the pollination treatment on the element concentrations and on the ratio between K and Ca (K : Ca) were analysed with separate linear mixed-effect (LME) models, from the nlme-package [34], for each element. Fixed factors were pollination treatment, initial weight and their interaction. Total buds per tree, final fruit set per branch, colour cover and site were included as covariates and treeID as a random effect. The continuous predictors were centred, using the scale-function. Response variables were transformed if needed to meet the assumption of normally distributed model residuals and the interaction term was deleted if non-significant. Models were evaluated using likelihood-ratio tests between full models and models with one term dropped. Extra apples and apples with missing values in the fixed factors or covariates were not included in analyses (resulting in $n = 190$). Two outliers were removed from the Ca analyses, owing to extremely high Ca values.

To analyse DMC of fruits in relation to treatment and storage time, the data were divided into two datasets. One dataset, healthy apples ($n = 153$), included fruits that were measured initially, directly after harvest, and fruits that persisted in the storage until the end of the experiment. The second dataset ($n = 37$) included apples that suffered from postharvest disorders during the storage time. LME models were fitted for both datasets, with per cent dry matter as the response variable, pollination treatment and storage category (initial and final) or days in storage, and their interactions, as the explanatory variables, and final fruit set per branch, initial fruit weight and total number of buds per tree as covariates. The factor site was included to account for differences between sites, and treeID as a random effect. The interaction term was deleted if non-significant and the models were further simplified with the drop1 function until the AIC no longer decreased. If a treatment effect was found, we performed post hoc tests using the glht-function with the same predefined contrast as above. The categorical variable 'storage category' was changed to the continuous variable 'days in storage' for the post hoc test. Continuous predictors were centred, using the scale-function and $p$-values were obtained using likelihood-ratio tests. The other quality variables, sugar content, firmness, titrated acidity and weight loss during storage, which are all highly inter-correlated, were analysed with similar models as the DMC, and included treatment, storage category/storage time and their interaction, fruit weight and site as predictor variables. The results for sugar content, firmness, acidity and weight loss are presented in the electronic supplementary material, table S2 and figure S3.

The relationship between storage duration, i.e. the number of days the fruit were in storage before suffering from postharvest disorders, and element concentrations in fruits, was analysed with the Cox proportional hazards regression models, with the coxph-function from the survival-package in R [39]. The Cox proportional hazards regression models are used for evaluating how specific variables (element concentrations) influence the rate of a particular event happening (post-storage disorders) at a particular point in time (referred to as the hazard rate). A ratio of the hazard rates (referred to as hazard ratio) greater than 1 indicates that a variable has a negative influence on storage duration. Survival status at the end of the experiment was included in the model, i.e. if the apples were still healthy or not. The weight of the apples in g and the interaction between weight and element concentrations were included as covariates in the initial model, but dropped if they did not improve the model fit. Site was included either as a covariable, a stratification variable or dropped depending on if it fulfilled the assumption that the hazard ratio was constant over time. Element concentrations were log-transformed when it improved model fit. The model assumption of proportional hazards was tested with the cox.zph function (survival-package) and by plotting the Schoenfeld residuals versus time, and the assumption of linear covariates was verified by plotting Martingale residuals against each covariate. $p$-values for elements and weight were obtained with Wald tests, where $z$ gives the Wald statistic.

To assess direct and indirect effects of the pollination treatment on the storability of apples, we developed a SEM. Before developing the model, we decided to include only the K : Ca ratio, and not the other elements, in the SEM, because it is well established that K : Ca ratios affect postharvest disorders and thus apple storability [24,25,40], and because our previous analyses indicated that the K : Ca ratio was related to both pollination treatment and increased risk for postharvest disorders (tables 1 and 2). We used individual LME to be included in an initial piecewise SEM, where all models had

**Table 1.** The effect of the interaction between pollination treatment and the apple weight (g) at harvest on the element concentrations, analysed with linear mixed-effect models (LME, nlme-package in R) with the random effect treeID ($n = 190$). When the interactions were non-significant, the effect of treatment and apple weight is presented separately. $p$-values were obtained using log-likelihood tests and are presented in bold when significant ($p < 0.05$). The predictor variables total buds per tree, final fruit set per branch, cover colour of the apples and site were also included in the LME and the $p$-values for these variables, obtained using log-likelihood tests, are reported in the supplementary material (electronic supplementary material, table S3).

| response variable | treatment × apple weight (d.f. = 2) | slope estimates ± s.e. | | | treatment (d.f. = 2) | apple weight (d.f. = 1) |
| --- | --- | --- | --- | --- | --- | --- |
| | | pollinator exclusion | natural pollination | supplementary hand-pollination | | |
| Log(K : Ca) | $p = 0.34$ | | | | $p = 0.057$ poll ex: 3.5 ± 0.10 con: 3.5 ± 0.09 supp: 3.3 ± 0.12 | $p = 0.069$ |
| Ca[a] | $p = 0.20$ | | | | $p = 0.10$ | **$p = 0.002$** |
| K | **$p = 0.025$** | −0.86 ± 0.42 | −1.06 ± 0.36 | 0.69 ± 0.58 | | |
| sqrt(Zn) | **$p = 0.012$** | −0.000054 ± 0.00029 | −0.0011 ± 0.00025 | −0.0011 ± 0.00042 | | |
| sqrt(P) | $p = 0.062$ | | | | $p = 0.16$ | $p = 0.17$ |
| log(Fe) | $p = 0.30$ | | | | $p = 0.14$ | $p = 0.99$ |
| Mg | **$p < 0.001$** | −0.071 ± 0.021 | −0.10 ± 0.018 | 0.052 ± 0.030 | | |
| log(B) | $p = 0.32$ | | | | $p = 0.24$ | $p = 0.19$ |

[a]Two outliers were removed.

**Table 2.** The effect of element concentration on storage duration analysed with Cox proportional hazards regression models (coxph-function, survival-package in R). The variables included in the final models are seen in the column 'variables'. $\beta$ is the regression coefficient where a positive sign represents greater risk of postharvest disorders. HR and 95% CI is the hazard ratio with its 95% confidence interval showing the effect size of the covariates. Bold $p$-values represent significant relationships ($p < 0.05$). The last column shows if site has been included as a co-variable, a stratification-variable or has been dropped from the model. A total of 145 apples were included in the models.

| element | variables | $\beta \pm$ s.e. | HR | 95% CI | z-statistic | p-value | site |
|---|---|---|---|---|---|---|---|
| K : Ca | log(K : Ca) * apple weight | $-0.012 \pm 0.006$ | 0.99 | 0.98–0.99 | −1.99 | **0.046** | strata |
| | log(K : Ca) | $2.740 \pm 1.092$ | 15.48 | 1.82–131.74 | 2.51 | **0.012** | |
| | apple weight | $0.048 \pm 0.023$ | 1.05 | 1.00–1.10 | 2.29 | **0.035** | |
| Ca[a] | Ca | $-0.019 \pm 0.013$ | 0.98 | 0.96–1.01 | 0.15 | 0.147 | strata |
| K | K | $0.0022 \pm 0.00089$ | 1.002 | 1.0004–1.00393 | 2.43 | **0.015** | co-variable |
| | apple weight | $0.0078 \pm 0.0032$ | 1.01 | 1.00–1.01 | 2.25 | **0.015** | |
| Zn | log(Zn) | $1.83 \pm 0.45$ | 6.25 | 2.58–15.14 | 4.06 | **<0.001** | co-variable |
| | apple weight | $0.011 \pm 0.0034$ | 1.01 | 1.00–1.02 | 3.2 | **0.001** | |
| P | P | $0.015 \pm 0.0084$ | 1.02 | 0.99–1.03 | 1.79 | 0.073 | co-variable |
| | apple weight | $0.0060 \pm 0.0031$ | 1.01 | 1.00–1.01 | 1.90 | 0.058 | |
| Fe | Fe * apple weight | $0.0045 \pm 0.0080$ | 1.00 | 0.99–1.02 | 0.57 | 0.569 | strata |
| | Fe | $-0.20 \pm 1.33$ | 0.82 | 0.060–11.06 | −0.15 | 0.878 | |
| | apple weight | $0.00036 \pm 0.0093$ | 1.00 | 0.98–1.02 | −0.038 | 0.969 | |
| Mg | Mg | $0.013 \pm 0.016$ | 1.01 | 0.98–1.00 | 0.79 | 0.243 | strata |
| | apple weight | $0.0063 \pm 0.0035$ | 1.07 | 1.0–1.01 | 1.80 | 0.072 | |
| B | B | $-0.13 \pm 0.11$ | 0.86 | 0.71–1.09 | −1.21 | 0.228 | strata |
| | apple weight | $0.0056 \pm 0.0033$ | 1.01 | 1.00–1.01 | 1.71 | 0.087 | |

[a]Two outliers were removed.

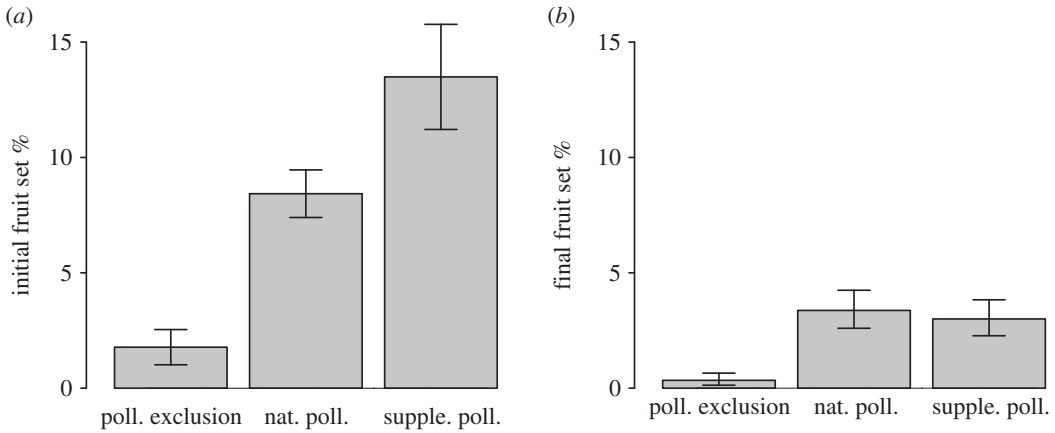

**Figure 1.** The effect of the treatments 'pollinator exclusion', 'natural pollination' and 'supplementary hand-pollination' on (*a*) the initial fruit set (number of initial fruits divided by number of flowers per branch) and (*b*) the final fruit set (number of ripe fruits divided by number of flowers per branch), using predicted values from the GLS models. Bars represent model-estimated standard errors. Estimated means per treatment and the standard errors in (*b*) are back-transformed from squared rooted values.

treeID as a random effect. Each model was first individually evaluated by plotting standardized residual against fitted values and against each predictor variable [41]. When necessary, we used the variance function varPower to model heteroscedasticity and log-transformation to achieve normality of residuals. To assess the model fit of the SEM, we used directional separation test (D-separation test) which yields Fisher's $C$-statistic that is $\chi^2$ distributed. If a missing path was detected with the D-separation test, the path was added. To simplify the model, insignificant paths were deleted from the model until the SEM's AIC no longer declined or no other paths could be deleted. The relative importance of continuous predictor variables was compared using standardized path coefficients.

# 3. Results

## 3.1. Fruit and seed set

The pollination treatment had an effect on both initial (L.ratio = 48.2, $\Delta$d.f. = 2, $p < 0.001$) and final fruit set (L.ratio = 39.4, $\Delta$d.f. = 2, $p < 0.001$) (figure 1). The naturally pollinated branches had 374% higher initial ($p < 0.001$) and 200% higher final ($p < 0.001$) fruit set than the branches in the pollinator exclusion treatment, which confirms that Amorosa apples are highly dependent on animal-mediated pollination. The supplementary hand-pollinated flowers had 60% higher initial ($p = 0.04$) fruit set compared to the naturally pollinated branches, indicating lower fertilization success in the natural compared to the supplementary hand-pollinated treatment. However, this difference was no longer found in the final fruit set when only considering the subset of flowers that were supplementary hand-pollinated ($p > 0.5$), but was still present when considering the fruit set of the entire branch ($p = 0.04$, electronic supplementary material, figure S1). The pollination treatment also affected the seed set in apples ($\chi^2 = 67.3$, $\Delta$d.f. = 2, $p < 0.001$), with higher seed set in naturally pollinated fruits (mean = 2.3) compared to fruits in the pollinator exclusion treatment (mean = 0.4, $p < 0.001$), while no difference was found between naturally (mean = 2.3) and supplementary hand-pollinated fruits (mean = 2.4, $p > 0.5$).

## 3.2. Mineral content in fruits

The concentrations of the elements K, Zn and Mg in fresh weight apples were affected by the interaction between pollination treatment and initial apple weight (table 1), indicating that the effect of pollination treatment on mineral content depended on the weight of the apples (table 1; electronic supplementary material, figure S2). For K and Mg, the interaction effect arose because the supplementary hand-pollinated apples increased in K and Mg concentrations with increased apple weight, whereas the naturally pollinated (K: $t = -2.6$, $p = 0.009$, Mg: $t = -4.5$, $p < 0.001$) and pollinator-excluded fruits (K: $t = -2.2$, $p = 0.03$, Mg: $t = -3.8$, $p = 0.001$) decreased in element concentrations with increased apple weight (table 1; electronic supplementary material, figure S2). The variation in Zn concentration arose because

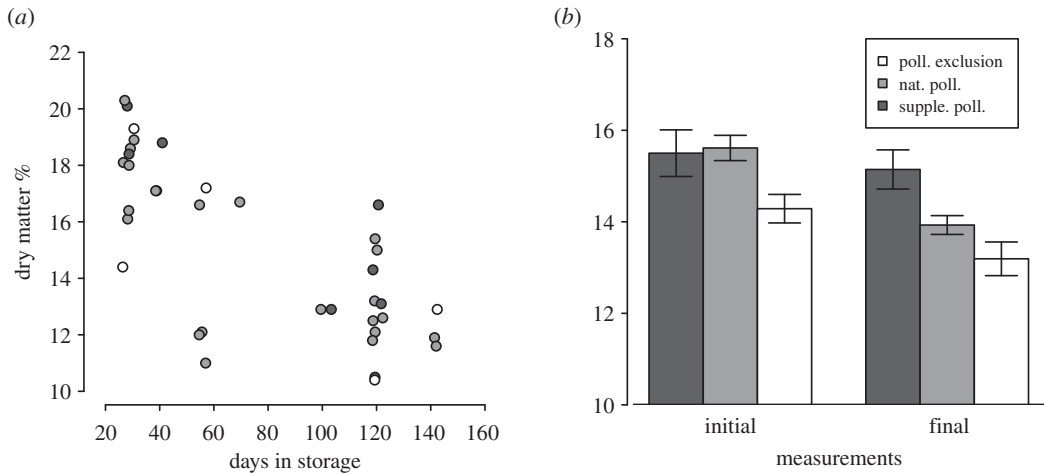

**Figure 2.** DMC (%) in relation to (*a*) the number of days the fruits were in storage before suffering from postharvest disorders (*n* = 37) and (*b*) healthy fruits from the categories initial (directly after harvest) and final measured fruits (after 161–162 days in storage) (*n* = 153). Apples that had started to shrivel were considered as healthy first-class fruit if no other damage was detected on the skin. Supplemented pollinated fruits are represented by dark grey, controls by grey and pollinator excluded by white. In (*a*), dots are jittered to make dots more visible.

control ($t = -2.9$, $p = 0.005$) and supplementary hand-pollinated fruits ($t = -2.1$, $p = 0.039$) had a steeper decline in Zn concentration with increased weight compared to pollinator excluded fruits. We found a marginally non-significant trend that pollinator treatment affected the K : Ca ratio in fruits (L.ratio = 5.7, $p = 0.057$), with supplementary hand-pollinated fruits tending to have lower K : Ca ratio compared to the other treatments (table 1). For the remaining elements (Ca, P, Fe and B), the concentrations did not vary between pollination treatments or as a consequence of an interaction between pollination treatment and apple weight (table 1). However, the Ca concentration was negatively related to the weight of the apples (table 1). The average concentrations for the measured elements in fresh weight and dry matter apple are presented in the electronic supplementary material, table S1.

## 3.3. The effect of storage time on fruit dry matter and mineral content

DMC in fruits was affected by pollination treatment in both healthy fruits (L.ratio = 27.8, Δd.f. = 2, $p < 0.001$ (initial and finally measured fruits)) and fruits that suffered from postharvest disorders during storage (L.ratio = 9.2, Δd.f. = 2, $p = 0.010$). Fruits from both health categories had higher DMC in the supplementary hand-pollinated treatment than in the natural pollination treatment (pairwise comparison natural pollination: $z = 4.0$, $p < 0.001$ (initial and finally measured fruits); $z = 2.56$, $p = 0.021$ (apples that suffered from postharvest disorders)), and naturally pollinated fruits had higher DMC compared to the pollinator-excluded fruits (pairwise comparison with pollinator-excluded fruits: $z = -2.9$, $p = 0.004$ (initial and finally measured fruits); $z = -2.2$, $p = 0.027$ (apples that suffered from postharvest disorders)) (raw data shown in figure 2). DMC decreased with days in storage in fruits that suffered from postharvest disorders (L.ratio = 20.9, Δd.f. = 1, $p < 0.001$). DMC also had a tendency to be higher in apples that were measured directly after harvest compared to apples that had remained in the storage for the entire storage period (greater than 160 days) (L.ratio = 3.6 Δd.f. = 1, $p = 0.059$). We found no interaction effect between pollination treatment and storage time in these analyses, showing that fruits from the supplementary hand-pollination treatment had consistently higher DMC throughout the storage time.

The concentration of the elements K, Zn and the K : Ca ratio affected the risk for apples to suffer from postharvest disorders during storage (analysed with the Cox proportional hazards regression models, table 2). Apples with higher K and Zn concentrations had higher risk of suffering from postharvest disorders during storage (table 2). An interaction between the K : Ca ratio and fruit weight also affected the risk of postharvest disorders, which means that the weight of the apples modified the risk with having higher K : Ca ratios. The risk of suffering from postharvest disorders with higher K : Ca ratio decreased with increasing fruit weight (table 2). The fruit weight variable was also significant in the models for K and Zn, indicating that higher fruit weight increased the risk of postharvest disorders (table 2). No other element concentrations were related to storage duration.

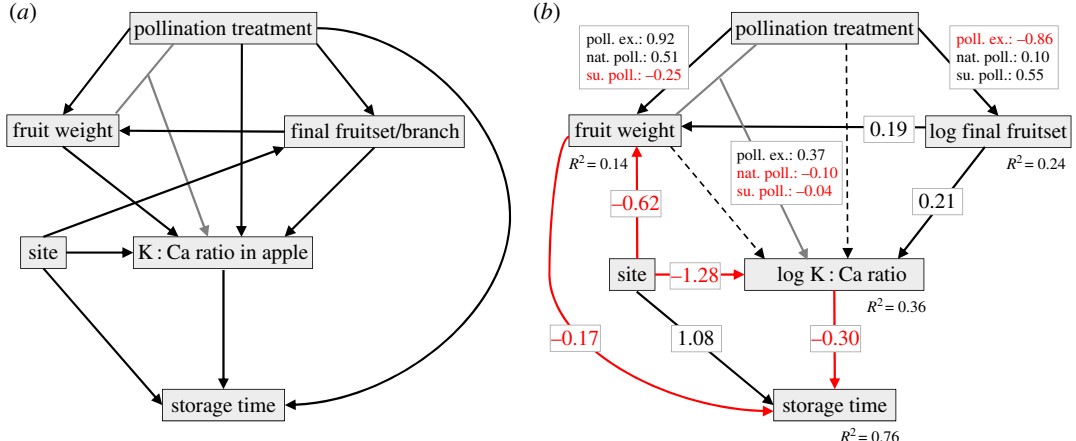

**Figure 3.** Piecewise SEM exploring the relationship among pollination treatment, fruit weight, fruit set, fresh weight K : Ca ratio, site and storage time. (*a*) Initial SEM showing all tested paths and (*b*) is the final analysed SEM after variable transformations, addition of a missing path and deletion of insignificant relationships. Solid arrows represent significant unidirectional relationships ($p < 0.05$), where black arrows represent a positive relationship, red arrows a negative relationship, grey arrow an interaction term and the dashed lines the variables included in the interaction term. The numbers in the boxes on top of the arrows represent the standardized path coefficients (for continuous variables); the numbers in the box on top of the interaction term represent the slope estimates for the different pollination treatments.

## 3.4. Structural equation model

When running the initial SEM (figure 3*a*) to explore the causal relationships between pollination treatment, fruit weight, final fruit set, K : Ca ratio and storage duration, the D-separation test revealed two missing links; one between fruit weight and storage duration and one between site and fruit weight. These paths were added to the model and non-significant paths were deleted (figure 3*b*), resulting in a model with better fit than the initial model (Fisher's $C = 8.9$, d.f. = 10, AIC = 66.9, $p = 0.54$ compared to initial: Fisher's $C = 18.2$, d.f. = 10, AIC = 78.2, $p = 0.051$). In the final SEM, we found that pollination treatment had an indirect effect on storage time of fruits, mediated through final fruit set, fruit weight, an interaction with fruit weight and the K : Ca ratio (figure 3*b*, see also $\chi^2$ likelihood-ratio test statistics; electronic supplementary material, table S4). Fruit weight had both a direct negative effect on storage time and an indirect interaction effect with pollination treatment that affected the K : Ca ratio, which in turn with increased ratio had a negative effect on storage time (figure 3*b*). The interaction effect between pollination treatment and weight arose because pollinator-excluded fruits increased in K : Ca ratio with increased fruit weight, which was not seen for the other treatments. Site had a strong direct effect on storage time as well as on the fruit weight and K : Ca ratio which in turn also affected storage time (figure 3*b*).

## 4. Discussion

The fruit quality aspects important for apple growers are those connected to marketability and storability. We found that the pollination treatment could affect both these aspects. The supplementary hand-pollination treatment, where compatible pollen had been added to the stigmas to increase the probability of cross-pollination, resulted not only in the highest initial fruit set, but also in higher DMC levels and a tendency for lower K : Ca ratio, compared with naturally pollinated apples. These quality aspects are desirable because previous studies suggest that higher DMC levels are connected to consumer preferred flavour, sucrose, acidity and fruit texture levels while lower K : Ca ratios are related to a reduced incidence of postharvest disorders [23,24]. Moreover, K and two other elements (Zn, Mg) were affected by an interaction between pollination treatment and fruit weight, indicating that the level of cross-pollination is affecting the accumulation of those elements into apple fruits. Our studies show that also a higher concentration of Zn, and not only K and higher K : Ca ratio, may influence the risk of postharvest disorders during storage.

The connections between pollination treatment and storability seem to be complex and indirect, going through several covariables. The SEM suggested that the pollination treatment had a direct effect on the

weight of apples, where fruits from the supplementary hand-pollinated treatment were the lightest and fruits from pollinator exclusions were the heaviest, which indirectly affected storability because lighter fruits had better storability. On the other hand, supplementary hand-pollination also increased final fruit set (on the entire branch level) which had a positive effect on the K : Ca ratio which in turn had a negative effect on storability. To complicate the matter further, the K : Ca ratio was affected by an interaction between pollination treatment and apple weight, where heavier apples from the pollinator-excluded treatment had higher K : Ca ratios compared to apples from the other pollination treatments. Hence, the weight of the apple modified the effect of pollination treatment on the K : Ca ratio, and therefore, it is not possible to evaluate which pollination treatment leads to longest storability without considering apple weight. However, because the supplementary hand-pollinated apples generally were lighter (figure 3), and tended to have a lower K : Ca ratio compared to other treatments (table 1), which are positive attributes for storability, this indicates that apples with ensured cross-pollination could have a generally better storability.

DMC, which is suggested to be a reliable predictive tool for estimating marketable fruit quality [16–18], was highest in apples that were supplementary hand-pollinated. Because higher DMC is related to better flavour [17] and increased consumer preference [16], high DMC levels could be a desirable trait for apple producers. In the fresh apple market, flesh firmness above a certain level together with high total soluble solids and TA are preferred and considered the main quality traits of fruits [7,16]. However, because most apples for consumption are harvested before fully ripe and because fruits consist of living tissues, the firmness, soluble solids and TA are not stable quality metrics but will change during maturation owing to metabolic activity [16]. Because fruits are harvested before starch solubilization is completed, the soluble solids at harvest do not represent the soluble solids after storage well [16]. Measurements of DMC at harvest have instead been found to better predict the total soluble solids in fruits after storage, probably because DMC also considers starch levels [16]. In our experiment, DMC was lower for fruits that remained longer in storage, which may be related to respiration during storage. Because we did not measure respiration, we cannot distinguish respirational effects on DMC from the possibility that fruits with high DMC suffered more from early postharvest disorders. On the other hand, the retained higher DMC throughout the storage time in supplemented hand-pollinated fruits indicated a persistently higher quality of ensured cross-pollinated fruits compared to the other treatments.

In contrast with earlier studies [23,24], we did not find that increased Ca content in apples by itself lowered the risk of postharvest disorders. However, an interaction between the K : Ca ratio and apple weight, and a higher Zn content affected the risk of postharvest disorders, where a high K : Ca ratio increased the risk of postharvest disorders mainly for lighter apples and less so for heavier apples. Corresponding to these results, the SEM showed an increased K : Ca ratio to have a negative impact on storability. Previous studies suggest that the low Ca levels in the K : Ca ratio rather than the high levels of K cause postharvest disorders [42] and that pollination affects the Ca concentration [29,30], and thus indirectly postharvest disorders. The mechanism suggested by Bramlage et al. [29] is that increased seed numbers induces higher auxin production which increase translocation of Ca into fruits, a mechanisms not found for other elements (K and Mg). However, in our study, we did not find Ca in fruits to be affected by pollination, but we found Mg and K levels to be affected by pollination treatment in interaction with fruit weight, which was not tested in the previous studies, and therefore possibly missed. While the seed number and auxin-related mechanism seems plausible, additional mechanisms may be at play as we found tendencies for differences in the K : Ca ratio in apples between control and supplementary hand-pollinated fruits but no differences in seed set.

Even though pollination treatment affected both quantity and several quality aspects of apples, other site factors also had a strong impact, as shown by the significant site effect in several of the analyses (e.g. figure 3; electronic supplementary material, table S3). Site effects may indicate the importance of unmeasured management practices like thinning and fertilization regimes, and site variation in the age of the apple trees, the availability of elements and nutrients in the soil, pests and pathogens and other animal interactions in the orchard, and light levels reaching the apples (e.g. [30,43,44]). These unmeasured factors may have concealed differences in final fruit set levels between natural and supplementary hand-pollinated fruits, while the effect of the pollinator treatments persisted for the quality variables. Hence, even considering site effects, increasing cross-pollination in orchards evidently needs to be considered in the management because it affected several quality aspects of apples. To increase successful cross-pollination from natural pollination, orchards need to have ample compatible pollen sources in configurations that facilitate pollinator movements between compatible cultivars [45]. In addition, high species richness and abundance of wild bees seem to be

the key [46–48]. Higher species richness of pollinators have generally higher functional trait diversity and can therefore complement each other both spatially and temporally, leading to increased pollination services [46,48,49]. Moreover, the effectiveness of managed bees as crop pollinators is enhanced by the presence of other interacting wild pollinators [50].

In order to increase sustainability of apple production and to understand the full economic effect of pollination services, we need a better understanding of the relationships between pollination services, cross-pollination and fertilization, fruit set, marketable fruit quality and storable fruit quality. In our study, we found increased amount of delivered compatible pollen to apple stigmas to be beneficial not only for fruit set but also for higher DMC in fruit. The link between pollination treatment and storability was more complicated and needs to consider the modifying effect of fruit weight on the effect of pollination treatment. However, lighter apples and lower $K:Ca$ ratio in supplementary hand-pollinated fruits indicate that these fruits have longer storability than fruits from flowers that had received less compatible pollen. Further research is needed to evaluate how general our results are and how they interact with other management decisions such as selection of cultivars, cultivar configuration in orchards, thinning-regimes and pollinator management.

Data accessibility. All data and code associated with this manuscript are available via the Dryad Digital Repository: https://datadryad.org/stash/dataset/doi:10.5061/dryad.nk9871p [51].

Authors' contributions. U.S. developed the method and project idea. U.S. collected and analysed the data. U.S., P.A.H. and H.G.S. interpreted the results and conceptualized the frame for the manuscript. U.S. wrote the first drafts of the manuscript. All authors contributed substantially to revisions and gave final approval for publication.

Competing interests. We declare we have no competing interests.

Funding. This study was supported by grants from Ekhagastiftelsen (to U.S.), the 2013–2014 BiodivERsA/FACCE-JPI joint call for research proposals (agreement no. BiodivERsA-FACCE2014-74), with the national funder FORMAS under grant no. 226-2014-1784 (to P.A.H.) and from Lund University (to H.G.S.). The research was part of the research environment BECC (Biodiversity and Ecosystem Services in a Changing Climate).

Acknowledgements. We are very grateful to Kiviks Musteri AB for their cooperation and the hard work by the field assistants Christine Sandberg and Lucy Seeger.

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
