## [Reviewer comments · Royal Society Open Science]

Review History

RSOS-190326.R0 (Original submission)

Review form: Reviewer 1

Is the manuscript scientifically sound in its present form?

Yes

Are the interpretations and conclusions justified by the results?

Yes

Is the language acceptable?

Yes

Is it clear how to access all supporting data?

Yes

Do you have any ethical concerns with this paper?

No

Have you any concerns about statistical analyses in this paper?

No

Recommendation?

Major revision is needed (please make suggestions in comments)

Comments to the Author(s)

Review of "Impact of pollination services to apples goes beyond increased fruit set"

Greetings editors and authors:

It always continues to amaze me that in such a well-studied crop like apple, there are still so many things we don't know about the reproductive biology of the plant, and the ultimate impacts on fruit quality. In this study, the authors investigate the impact of different pollination treatments (bagged flowers = no pollinators access; open pollination; and hand pollination with hand collected compatible pollen), on several metrics of fruit quantity (i.e., fruit set) and quality (i.e., fruit weight, element content, dry matter content (DMC) and K:Ca ratio). For some of these variables seem to have little bearing on fresh fruit marketing, they do indeed play a big role in the post-harvest physiology of the fruit, and are critical for extending the storability of the fruit which is important for having apples available to markets throughout the year. Thus, I think this manuscript will ultimately be of interest to many. I also think that it is very well-written, and my main concerns for its publication are only with the use of terms.

My biggest concern about this manuscript is that the authors use terminology associated with the word "pollination" incorrectly. To be as simple as possible, pollination simply refers to the delivery of pollen from anthers to stigmas, nothing more, nothing less. However, this means that one can have an apple flower that is adequately pollinated, but which produces no fruit, a misshaped fruit, or an excellent quality fruit – thus fruit quality, including the variables studied here may be more related to post-pollination events (i.e., fertilization) resulting from pollination with compatible pollen. Remember pollination can also occur with non-compatible pollen, and even self-pollination is still pollination, even though it may not produce fruit.

Throughout the manuscript, the authors use the term "pollination levels" which is not correct; this term should be used for when one assessed the actual number of pollen grains on the stigmas, which was not done here. This is not trivial; in one of my previous studies (Sheffield et al. 2005) I demonstrated that for many apple cultivars, the notion that pollen deposition needs to be heavy, and evenly distributed on the five stigmas of a flower is not necessarily true. Pollination can occur on only one of the five stigmatic surfaces to produce a good quality fruit due to an internal connection in the apple flower. As such, pollination does not have to be full, in fact, in theory five to ten compatible pollen grains on one stigmatic surface could produce a well-shaped fruit with a full complement of seeds. However, most flowers will receive 100s of pollen grains, though this is influenced by weather, and stage of flowering (see Sheffield 2014). The number of seeds does not have to be maximized, so the relationship between seed number and fruit size and shape is not always easy to interpret. In another work I (Sheffield 2014) showed that seed distribution, and not necessarily number was most important for fruit quality. I am happy to share these works if the authors wish to see them.

Thus, as an alternative to "pollination levels" I suggest that the authors use "pollination treatments" consistently throughout, the treatments being the control (bagged flowers), natural pollination, and hand pollination with compatible pollen. As such, you should also discuss that the "natural pollination" by bees was not necessarily the issue, but rather the pollen they were delivering itself. Pollinators don't care about the source of the pollen, so may be moving around compatible or non-compatible pollen; depending on the cultivar mix, distribution, and

overlapping flowering phenology, the bees may or may not be moving around suitable pollen, but that does not mean they are not adequately pollinating the blossoms. Thus the authors should be stressing pollination treatments having the impact (due to pollen compatibility, likely being better in the hand pollination treatments), not pollination levels, as the latter was not investigated. As such, the pollinators are likely doing a great job, and it may be that other factors are involved.

In this context, I think the authors should re-think their results based on pollen compatibility, not pollination levels, with respect to the metrics used to assess fruit quality. Remember that not all pollen is equal; studies in Canada by Kron et al. (cited in papers below) determined that competition between pollen types on a stigma occurs internally, so the desired impact on fruit quality via higher DMC and lower K:Ca ratio may be the result of the types of donor pollen(s), not the presence of the pollen itself (so, not necessarily a pollination issue). I think then the differences observed are not pollinator related, but linked to orchard design, cultivar mix and orchard layout, and pollinator movement patterns (they move down rows). Thus (and for example), from line 374 "...indicates that supplementary hand-pollinated apples could have a generally better storability" is only true in the context of the pollen, not pollination per say. You cannot state that pollination is better at all, as you did not look at that. All you can confirm is that when ample compatible pollen is delivered, then storability is better. This should not be seeming to suggest that bees are not performing.

Otherwise, I enjoyed this research, and hope the authors will find these comments (which are also on the attached manuscript (Appendix A)) useful.

Review form: Reviewer 2

Is the manuscript scientifically sound in its present form?

Yes

Are the interpretations and conclusions justified by the results?

Yes

Is the language acceptable?

Yes

Is it clear how to access all supporting data?

Yes

Do you have any ethical concerns with this paper?

No

Have you any concerns about statistical analyses in this paper?

No

Recommendation?

Accept with minor revision (please list in comments)

Comments to the Author(s)

This manuscript explores the importance of pollination for fruit production in commercial apple orchard. It extends previous research by asking how the level of pollination influencing the dry matter weight and mineral composition of the apples, rather than simply focusing on fruit and seed set. The results show that additional hand pollination increase the dry matter content and reduced the K/Ca ratio of fruits, components which are positively associated with greater storability and consumer preference.

Structural equation models were used to demonstrate that pollination influences the storability of apples through indirect mechanisms involving the mineral uptake into the fruits, a process that varies with pollination treatment and weight.

The study highlights that there is more to learn about the benefits that effective pollination can provide for commercial fruit growers.

The methods seem robust and have been analysed with appropriate statistical models. In places the manuscript would benefit from being made more concise, particularly in the discussion which has a tendency to repeat the results. I have a few queries about the methodology and minor suggestions which are listed below.

Abstract

Line 19 change increase to “can increase” or “increases”. Same with influence in the following line.

Methods

Line 138. You only hand pollinated 5 or so flowers per branch. It is clear that you only collected and weighed the fruits from the hand-pollinated flowers, but when calculating fruit set it sounds as if you counted all flowers and fruits on the branch. This seems inappropriate as the majority of flowers on the branch were not hand-pollinated so are equivalent to the control. Please could you clarify this.

Line 184 – Why did you exclude apples that did not meet the marketing standards? In line with your working hypothesis you might expect under-pollinated fruits to fall below the standards more frequently than those in the hand-pollination treatments so you seem to be excluding useful data without a clear explanation.

Line 213 – Please could you clarify what “possible dilution effect” and “initial investment by tree” are.

Line 247 I think hazard rate should be replaced by hazard ratio?

Results

Line 280. Again, am I right in thinking only 5 flowers per branch received additional hand-pollination, but that fruit set was calculated for the entire branch? This does not seem appropriate and be at least be justified and discussed.

Line 312 This [paragraph] would benefit from rewording. Currently the explanation of the post hoc test results is unclear and a bit confusing.

332. I think it is useful to use Structural Equation Model rather than SEM in the sub-title as other authors some times use this acronym for standard error of the mean.

Discussion

Line 359 Influence the risk in which direction?

Line 372. I suggest that you avoid referencing Figures and tables in the discussion. They are best kept to the results section.

Line 387 Found to predict it better than what?

The end of this paragraph could be made more concise. At the moment it is challenging to follow.

Line 395 found should be changed to find

Line 399 Parts of your discussion seem to be repeating the results section would benefit from being condensed, with the focus shifted towards interpreting rather than repeating the results

Line 406 Change found to find.

Figures and Tables are all clear and well presented, but it is not necessary to provide so much detail about the statistical methods in the figure legends as this information is better suited in the methods.

Decision letter (RSOS-190326.R0)

31-Jul-2019

Dear Dr Samnegård,

The editors assigned to your paper ("Impact of pollination services to apples goes beyond increased fruit set") have now received comments from reviewers. We would like you to revise your paper in accordance with the referee and Associate Editor suggestions which can be found below (not including confidential reports to the Editor). Please note this decision does not guarantee eventual acceptance.

Please submit a copy of your revised paper before 23-Aug-2019. Please note that the revision deadline will expire at 00.00am on this date. If we do not hear from you within this time then it will be assumed that the paper has been withdrawn. In exceptional circumstances, extensions may be possible if agreed with the Editorial Office in advance. We do not allow multiple rounds of revision so we urge you to make every effort to fully address all of the comments at this stage. If deemed necessary by the Editors, your manuscript will be sent back to one or more of the original reviewers for assessment. If the original reviewers are not available, we may invite new reviewers.

- Data accessibility

<http://datadryad.org/submit?journalID=RSOS&manu=RSOS-190326>

- Competing interests

- Authors' contributions

- Acknowledgements

- Funding statement

on behalf of Dr Stephen Long (Associate Editor) and Kevin Padian (Subject Editor)
openscience@royalsociety.org

Subject Editor Comments to Authors:

Thanks for your submission. The reviewers found it very interesting, but also raised some issues that should be addressed, including the use of terminology and the study methods. Please address these specifically in your revisions.

Also, could I suggest altering the title somewhat? I found it difficult to interpret, and for a general biological journal like RSOS readers should not have to struggle. Maybe something like "Pollination increases more than fruit set rates in commercial apples." You could even add a colon and in a sub-title list a few of the other outcome variables that are affected. Thanks for considering, and best wishes with your revisions.

Reviewers' Comments to Author:
Reviewer: 1

Review of "Impact of pollination services to apples goes beyond increased fruit set"

Greetings editors and authors:

It always continues to amaze me that in such a well-studied crop like apple, there are still so many things we don't know about the reproductive biology of the plant, and the ultimate impacts on fruit quality. In this study, the authors investigate the impact of different pollination treatments (bagged flowers = no pollinators access; open pollination; and hand pollination with hand collected compatible pollen), on several metrics of fruit quantity (i.e., fruit set) and quality (i.e., fruit weight, element content, dry matter content (DMC) and K:Ca ratio). For some of these variables seem to have little bearing on fresh fruit marketing, they do indeed play a big role in the post-harvest physiology of the fruit, and are critical for extending the storability of the fruit which is important for having apples available to markets throughout the year. Thus, I think this manuscript will ultimately be of interest to many. I also think that it is very well-written, and my main concerns for its publication are only with the use of terms.

My biggest concern about this manuscript is that the authors use terminology associated with the word "pollination" incorrectly. To be as simple as possible, pollination simply refers to the delivery of pollen from anthers to stigmas, nothing more, nothing less. However, this means that one can have an apple flower that is adequately pollinated, but which produces no fruit, a misshaped fruit, or an excellent quality fruit – thus fruit quality, including the variables studied here may be more related to post-pollination events (i.e., fertilization) resulting from pollination with compatible pollen. Remember pollination can also occur with non-compatible pollen, and even self-pollination is still pollination, even though it may not produce fruit.

Throughout the manuscript, the authors use the term “pollination levels” which is not correct; this term should be used for when one assessed the actual number of pollen grains on the stigmas, which was not done here. This is not trivial; in one of my previous studies (Sheffield et al. 2005) I demonstrated that for many apple cultivars, the notion that pollen deposition needs to be heavy, and evenly distributed on the five stigmas of a flower is not necessarily true. Pollination can occur on only one of the five stigmatic surfaces to produce a good quality fruit due to an internal connection in the apple flower. As such, pollination does not have to be full, in fact, in theory five to ten compatible pollen grains on one stigmatic surface could produce a well-shaped fruit with a full complement of seeds. However, most flowers will receive 100s of pollen grains, though this is influenced by weather, and stage of flowering (see Sheffield 2014). The number of seeds does not have to be maximized, so the relationship between seed number and fruit size and shape is not always easy to interpret. In another work I (Sheffield 2014) showed that seed distribution, and not necessarily number was most important for fruit quality. I am happy to share these works if the authors wish to see them.

Thus, as an alternative to “pollination levels” I suggest that the authors use “pollination treatments” consistently throughout, the treatments being the control (bagged flowers), natural pollination, and hand pollination with compatible pollen. As such, you should also discuss that the “natural pollination” by bees was not necessarily the issue, but rather the pollen they were delivering itself. Pollinators don’t care about the source of the pollen, so may be moving around compatible or non-compatible pollen; depending on the cultivar mix, distribution, and overlapping flowering phenology, the bees may or may not be moving around suitable pollen, but that does not mean they are not adequately pollinating the blossoms. Thus the authors should be stressing pollination treatments having the impact (due to pollen compatibility, likely being better in the hand pollination treatments), not pollination levels, as the latter was not investigated. As such, the pollinators are likely doing a great job, and it may be that other factors are involved.

In this context, I think the authors should re-think their results based on pollen compatibility, not pollination levels, with respect to the metrics used to assess fruit quality. Remember that not all pollen is equal; studies in Canada by Kron et al. (cited in papers below) determined that competition between pollen types on a stigma occurs internally, so the desired impact on fruit quality via higher DMC and lower K:Ca ratio may be the result of the types of donor pollen(s), not the presence of the pollen itself (so, not necessarily a pollination issue). I think then the differences observed are not pollinator related, but linked to orchard design, cultivar mix and orchard layout, and pollinator movement patterns (they move down rows). Thus (and for example), from line 374 “...indicates that supplementary hand-pollinated apples could have a generally better storability” is only true in the context of the pollen, not pollination per say. You cannot state that pollination is better at all, as you did not look at that. All you can confirm is that when ample compatible pollen is delivered, then storability is better. This should not be seeming to suggest that bees are not performing.

Otherwise, I enjoyed this research, and hope the authors will find these comments (which are also on the attached manuscript) useful.

Reviewer: 2

Comments to the Author(s)

This manuscript explores the importance of pollination for fruit production in commercial apple orchard. It extends previous research by asking how the level of pollination influencing the dry matter weight and mineral composition of the apples, rather than simply focusing on fruit and seed set. The results show that additional hand pollination increase the dry matter content and reduced the K/Ca ratio of fruits, components which are positively associated with greater storability and consumer preference.

Structural equation models were used to demonstrate that pollination influences the storability of apples through indirect mechanisms involving the mineral uptake into the fruits, a process that varies with pollination treatment and weight.

The study highlights that there is more to learn about the benefits that effective pollination can provide for commercial fruit growers.

The methods seem robust and have been analysed with appropriate statistical models. In places the manuscript would benefit from being made more concise, particularly in the discussion which has a tendency to repeat the results. I have a few queries about the methodology and minor suggestions which are listed below.

Abstract

Line 19 change increase to “can increase” or “increases”. Same with influence in the following line.

Methods

Line 138. You only hand pollinated 5 or so flowers per branch. It is clear that you only collected and weighed the fruits from the hand-pollinated flowers, but when calculating fruit set it sounds as if you counted all flowers and fruits on the branch. This seems inappropriate as the majority of flowers on the branch were not hand-pollinated so are equivalent to the control. Please could you clarify this.

Line 184 – Why did you exclude apples that did not meet the marketing standards? In line with your working hypothesis you might expect under-pollinated fruits to fall below the standards more frequently than those in the hand-pollination treatments so you seem to be excluding useful data without a clear explanation.

Line 213 – Please could you clarify what “possible dilution effect” and “initial investment by tree” are.

Line 247 I think hazard rate should be replaced by hazard ratio?

Results

Line 280. Again, am I right in thinking only 5 flowers per branch received additional hand-pollination, but that fruit set was calculated for the entire branch? This does not seem appropriate and be at least be justified and discussed.

Line 312 This [paragraph would benefit from rewording. Currently the explanation of the post hoc test results is unclear and a bit confusing.

332. I think it is useful to use Structural Equation Model rather than SEM in the sub-title as other authors some times use this acronym for standard error of the mean.

Discussion

Line 359 Influence the risk in which direction?

Line 372. I suggest that you avoid referencing Figures and tables in the discussion. They are best kept to the results section.

Line 387 Found to predict it better than what?

The end of this paragraph could be made more concise. At the moment it is challenging to follow.

Line 395 found should be changed to find

Line 399 Parts of your discussion seem to be repeating the results section would benefit from being condensed, with the focus shifted towards interpreting rather than repeating the results

Line 406 Change found to find.

Figures and Tables are all clear and well presented, but it is not necessary to provide so much detail about the statistical methods in the figure legends as this information is better suited in the methods.

Author's Response to Decision Letter for (RSOS-190326.R0)

See Appendix B.

RSOS-190326.R1 (Revision)

Review form: Reviewer 1

Is the manuscript scientifically sound in its present form?

Yes

Are the interpretations and conclusions justified by the results?

Yes

Is the language acceptable?

Yes

Do you have any ethical concerns with this paper?

No

Have you any concerns about statistical analyses in this paper?

No

Recommendation?

Accept as is

Comments to the Author(s)

Dear authors,

I find this version of your manuscript to be very good, and am happy with the effort you put into addressing my previous comments, and those of the other reviewer! As before, I feel that this is a well-written work, and have no additional comments.

As indicated previously, I find this study well worth publication; for crops and crop pollination, we seldom hear of the secondary impacts (not fruit quantity and shape) that adequate pollination/fertilization have on other aspects of fruit quality, even though we know that these are very important for consumer choice, fruit storage and longevity of quality, and aspects of health and diet. I think the fact that supplemental pollination was overall better than natural pollination is also telling of orchard design, cultivar mix, and pollinator behaviour. Thus, I think this work will be an important addition to furthering our understanding of fruit production in apple. Well done!

Decision letter (RSOS-190326.R1)

13-Nov-2019

Dear Dr Samnegård,

It is a pleasure to accept your manuscript entitled "Pollination treatment affects fruit set and modifies marketable and storable fruit quality of commercial apples" in its current form for publication in Royal Society Open Science. The comments of the reviewer(s) who reviewed your manuscript are included at the foot of this letter.

on behalf of Prof Kevin Padian (Subject Editor)
openscience@royalsociety.org

Associate Editor Comments to Author:

Thank you kindly for submitting your revision. The referee is happy with the author's response to their comments, and has no additional comments to be made.

Reviewer comments to Author:

Reviewer: 1

Comments to the Author(s)

Dear authors,

I find this version of your manuscript to be very good, and am happy with the effort you put into addressing my previous comments, and those of the other reviewer! As before, I feel that this is a well-written work, and have no additional comments.

As indicated previously, I find this study well worth publication; for crops and crop pollination, we seldom hear of the secondary impacts (not fruit quantity and shape) that adequate pollination/fertilization have on other aspects of fruit quality, even though we know that these are very important for consumer choice, fruit storage and longevity of quality, and aspects of health and diet. I think the fact that supplemental pollination was overall better than natural pollination is also telling of orchard design, cultivar mix, and pollinator behaviour. Thus, I think this work will be an important addition to furthering our understanding of fruit production in apple. Well done!

Appendix A**ROYAL SOCIETY
OPEN SCIENCE****Impact of pollination services to apples goes beyond
increased fruit set**

Journal:	Royal Society Open Science
Manuscript ID	RSOS-190326
Article Type:	Research
Date Submitted by the Author:	24-Feb-2019
Complete List of Authors:	Samnegård, Ulrika; Lunds Universitet, Department of Biology Hambäck, Peter; Stockholm University, Ecology, Environment and Plant Sciences Smith, Henrik; Lund University, Biology
Subject:	ecology < BIOLOGY, plant science < BIOLOGY, health and disease and epidemiology < BIOLOGY
Keywords:	dry matter content, Malus domestica, minerals, pollination, fruit quality, storage time
Subject Category:	Biology (whole organism)

**1 Impact of pollination services to apples goes beyond increased fruit set**

Ulrika Samnegård^{1,2*}, Peter A. Hambäck³ & Henrik G. Smith^{1,2}

¹Centre for Environmental and Climate Research, Lund University, SE-223 62 Lund, Sweden.

²Department of Biology, Lund University, SE-223 62 Lund, Sweden.

³Department of Ecology, Environment and Plant Sciences, Stockholm University, SE-106 91

Stockholm, Sweden.

*Corresponding author: Ulrika Samnegård, email: ulrika.samnegard@biol.lu.se,

ulrika.samnegard@gmail.com

Present address: Ulrika Samnegård, School of Environmental and Rural Sciences, University of New

England, Armidale, NSW, Australia

**18 Abstract**

**19** Insect-mediated pollination increase yields of many crop species and some evidence suggests that it
**20** also influence crop quality. However, the mechanistic linkages between insect-mediated pollination
**21** and crop quality are poorly known. In this study, we explored how pollination levels affected fruit set,
**22** dry matter content (DMC), mineral content and storability of apples. The highest pollination level
**23** (achieved through supplementary hand-pollination), resulted in higher fruit set, higher DMC and a
**24** tendency for lower K:Ca ratio, with the latter two being desirable quality aspects since higher DMC is
**25** connected to higher consumer preference and lower K:Ca ratio is related to lower incidence of
**26** postharvest disorders during storage. Using structural equation modelling, we showed an indirect
**27** effect of pollination level on storability, however mediated by complex interactions between fruit set,
**28** fruit weight and K:Ca ratio. The concentration of several elements in apples (K, Zn, Mg) was affected
**29** by the interaction between pollination treatment and apple weight, indicating that pollination level
**30** effects element allocation into fruits. In conclusion, our study shows that pollination level needs to be
**31** considered in the management of orchard systems, not only to increase fruit set, but also to increase
**32** the quality and potentially the storability of apples.

**34 Keywords:** dry matter content, fruit quality, *Malus domestica*, minerals, pollination, storage time

Introduction

Pollination services increase fruit and seed set of many crops, in particular that of vegetables, fruits,
berries and nuts [1]. For this reason, it is well-recognized that animal-mediated pollination is important
for the global food production in general and human nutrition in particular [1-3]. However,
accumulating evidence shows that the positive effect of pollination services goes beyond increasing
fruit and seed set, and may also increase quality [rapeseed, strawberries; 4, 5], affect nutritional
composition [almonds, apples; 6, 7], decrease malformations [Fuji apples, strawberries; 8, 9] and
increase shelf life-time [strawberries; 4, 9] of crops. However, the linkages between pollination
services and fruit quality have only just started to be recognized and there is a need for research that
simultaneously evaluate several quality aspects and their interrelated effects.

Apple is a fruit crop with a strong dependence on animal-mediated pollination, in which all cultivars
to some extent are self-incompatible and therefore require pollen transfer from another pollinizer
cultivar to set commercially acceptable fruit levels [10]. It is the most geographically wide-spread
temperate fruit [11] and the most common pollinator-dependent crop in Europe, where economic gains
from pollination-induced increases in fruit set is higher than those of any other crop [12]. However, for
commercially produced fruits not only the quantity of fruit matter nce marketable fruits also need
to be of adequate quality for good storability and to attract consumers. Even though higher insect
ollination levels e known to improve fruit yields of apples [13, 14], the influence of insect
pollination on quality aspects are more equivocal [7].

Several quality aspects have large economic impact on he e production. Important quality
attributes for consumers include flavour and flesh firmness rmer fruits are considered to have higher
quality by consumers [15, 16]. Flavour is a complex attribute that is related to the dry matter content
(DMC) in fruits, where higher DMC generally increases consumer preference [16, 17]. The DMC also
influences the firmness of apples at harvest and softening rates during storage, and is a good estimate

of total soluble solids after storage [16, 18, 19]. Another important quality aspect for producers and
wholesalers is storability. Apples can be stored for protracted periods of time, which allows for longer
market availability. However, even though the storage facilities have developed substantially, which
has prolonged the storability, much fruit is still discarded when taken out from storage. For example,
in Sweden, 20% of organic apples were disregarded at an experiment in 2010 and, depending on
cultivar, 9 – 27% of conventionally produced apples from seven orchards were disregarded after
storage during 2010-2015 (Ibrahim Tahir, pers. comm.). Many aspects influence the storability of
apples, where harvest time and mineral concentrations are important modifiers. For example, low
calcium content, and the high ratio between magnesium or potassium and calcium, is connected to
postharvest disorders including bitterpit, lenticel breakdown and Jonathan spot [20-24]. Calcium and
its ratio with other elements (e.g. K:Ca and N:Ca) is also connected to the softening of apples and
resistance to diseases [25-27]. Consequently, calcium application both before and after harvest to
increase fruit Ca-content is a common management action in modern orchards [22, 26].

A few studies have suggested that well-pollinated apples differ in their mineral content from less
pollinated ones, suggesting that the mechanisms influencing mineral allocation into fruits is related to
pollination services. Porcel *et al.* [28] found a positive relationship between seed number and calcium,
potassium and magnesium content in Aroma apples, Bramlage *et al.* [29] found a positive relationship
between seed number and calcium concentrations in Richared Delicious apples, and Volz *et al.* [30]
found that supplementary pollination positively affected calcium concentrations of Braeburn apples.
Other data suggest that the effect of pollination on calcium concentrations may be cultivar specific.
Bucceri and Di Vaio [31] found a positive relationship between higher seed set and calcium
concentration in fruits from some cultivars (Red and Golden Delicious) but not from other (Annurca
Rossa del Sud and Annurca Tradizionale), and Garratt *et al.* [7] found that supplementary hand-
pollination even decreased calcium concentrations in Gala apples. Since the mineral content may be
related to other quality aspects including firmness, postharvest disorders and storability, pollination

service may have a more far-reaching role in the economy of apple production than has earlier been
estimated.

The aim of this study was to simultaneously evaluate the direct effects from different levels of
pollination on the mineral concentration in apples, the marketable fruit quality of apples and the
indirect effects on the storable fruit quality. As an estimate of marketable fruit quality, we used DMC
as an endpoint variable [cf. 16], and as estimates of storable fruit quality we used both the probability
of developing various storage disorders and the time that the fruits could be stored and still maintain
good quality. Our working hypothesis was that higher pollination services lead to higher element
concentrations in fruits, higher DMC and that well-pollinated fruits have a better storability. To
examine this hypothesis, we both increased and decreased the pollen amount in apple flowers
compared to natural pollination through supplemental hand-pollination and through pollinator
exclusion, respectively. We also used a structural equation model to disentangle the direct and indirect
effects of pollination on storability, and particularly the K:Ca ratio which had previously been
implicated as a measure of storable fruit quality.

39 104 **Methods**

41 42 105 *Sites*

[revised manuscript text omitted]

Differences between pollination treatments for both initial and final fruit set were analysed by fitting
generalized least squares (GLS) models, in the nlme-package [34], using R [35]. Percent initial and
final fruit set was calculated by dividing the number of fruitlets and ripe fruits respectively with the
number of initial flowers on each branch and multiply it with 100. Percent initial and final fruit set
were included as response variables, and treatment and site as fixed factors. Final fruit set was square
root transformed to meet the assumption of normally distributed model residuals. For the initial fruit
set, a model allowing for unequal variance between pollination treatments was used (*VarIdent* option)
since it had better fit ($\Delta AIC = 27.7$). If a significant treatment effect was found using a likelihood-ratio
test, we performed posthoc tests using the *glht*-function from the “multcom package”, with Holm
adjusted p-values, in R [36] with the predefined contrasts control – supplementary hand-pollination
treatment and control – pollinator exclusion treatment. To make the *glht*-function to work with a GLS
model, an extra function from [http://rstudio-pubs-](http://rstudio-pubs-static.s3.amazonaws.com/13472_0daab9a778f24d3dbf38d808952455ce.html)
[static.s3.amazonaws.com/13472_0daab9a778f24d3dbf38d808952455ce.html](http://rstudio-pubs-static.s3.amazonaws.com/13472_0daab9a778f24d3dbf38d808952455ce.html), downloaded 2018-01-
[09](http://rstudio-pubs-static.s3.amazonaws.com/13472_0daab9a778f24d3dbf38d808952455ce.html), was used.

Pollination effects on seed set were analysed by fitting a generalized linear mixed-effects model, in the
lme4 package [37], with the binomial response variable seed set (maximum 10 developed seeds per
fruit), treatment as fixed factor and apple treeID as random effect. An observation-level random effect
was added to account for overdispersion [38]. If there was a significant treatment effect, we performed
posthoc tests as above. Extra apples were not included in the analysis and one sample lacked seed
count (n = 191).

The effects of the pollination treatment on the element concentrations and on the ratio between K and
Ca (K:Ca) were analysed with separate linear mixed effect models (*lme*), from the nlme-package [34],
for each element. Fixed factors were pollination treatment, initial weight (possible dilution effect), and
their interaction. Total buds per tree (initial investment by the tree), final fruit set per branch (possible

[revised manuscript text omitted]

indicating that the pollination level was involved in their accumulation into apple fruits (table 1). Our
studies show that also the Zn concentration, and not only K and the K:Ca ratio, may influence the risk
of postharvest disorders during storage (table 2).

The connections between pollination and storability seem to be complex and indirect, going through
several co-variables (figure 3). The SEM suggested that the pollination treatment had a direct effect on
the weight of apples, where higher pollination levels led to lighter fruits, which indirectly affected
storability because lighter fruits had better storability. On the other hand, higher pollination levels also
increased fruit set which had a positive effect on the K:Ca ratio which in turn had a negative effect on
storability. To complicated the matter further, the K:Ca ratio was affected by an interaction between
pollination treatment and apple weight, where heavier apples from the pollinator excluded treatment
had higher K:Ca ratios compared to apples from the other pollination treatments. Hence, the weight of
the apple modified the effect of pollination level on the K:Ca ratio and therefore it is not possible to
evaluate which pollination treatment that leads to longest storability without considering apple weight.

However, since the supplementary hand-pollinated apples generally were lighter (figure 3), and tended
to have lower K:Ca ratio compared to other treatments (table 1), which are positive attributes for
storability, this indicates that supplementary hand-pollinated apples could have a generally better
storability.

DMC, which is suggested to be a reliable predictive tool for estimating marketable fruit quality [16-
18], was highest in apples that were supplementary hand-pollinated. Since higher DMC is related to
better flavour [17] and increased consumer preference [16], high DMC levels could be a desirable trait
for apple producers. In the fresh apple market, flesh firmness above a certain level together with high
total soluble solids and TA are preferred and considered the main quality traits of fruits [7, 16].
However, since most apples for consumption are harvested before fully ripened and because fruits
consists of living tissues, the firmness, soluble solids and TA are not stable quality metrics but will
change during maturation due to metabolic activity [16]. Since fruits are harvested before starch
solubilisation is completed, the soluble solids at harvest do not represent the soluble solids after
storage well [16]. However, measurements of DMC at harvest, which mainly consist of soluble solids,
starch and organic acids in apples, have been found to better predict the total soluble solids in fruits
after storage, probably since it also consider starch levels [16]. In our experiment, DMC was lower for
fruits that remained longer in storage, which is not unexpected because of respiration during storage.
Since we did not measure respiration, we cannot distinguish this effect on DMC from the possibility
that fruits with high DMC suffered more from early postharvest disorders. On the other hand, we
found supplemented hand-pollinated fruits to retain higher DMC compared to the fruits from the other
treatments throughout the storage time, which indicates a persistently higher quality of well-pollinated
fruits.

In contrast to earlier studies [23, 24], we did not found that increased Ca content in apples by itself
lowered the risk of postharvest disorders (table 2). However, an interaction between the K:Ca ratio and
apple weight, and a higher Zn content affected the risk of postharvest disorders (table 2), where a high

399 K:Ca ratio increased the risk of postharvest disorders mainly for lighter apples and less so for heavier
apples. Corresponding to this results, the SEM showed an increased K:Ca ratio to have a negative
impact on storability (figure 3). Previous studies suggest that the low Ca levels in the K:Ca ratio rather
than the high levels of K cause postharvest disorders [42], and that pollination affects the Ca
concentration [29, 30], and thus indirectly postharvest disorders. The mechanism suggested by
Bramlage *et al.* [29] is that increased seed numbers induces higher auxin production which increase
translocation of Ca into fruits, a mechanisms not found for other elements (K and Mg). However, in
our study, we did not found Ca in fruits to be affected by pollination, but we found Mg and K levels to
be affected by pollination treatment in interaction with fruit weight, which was not tested in the
previous studies, and therefore possibly missed. While the seed number and auxin related mechanism
seems plausible, additional mechanisms may be at play as we found tendencies for differences in the
410 K:Ca ratio in apples between control and supplementary hand-pollinated fruits but no differences in
seed set.

Even though **pollination levels** affected several apple quality aspects, other site factors also had a
strong impact, as shown by the significant site effect in several of the analyses (e.g. figure 3, ESM
table S3). Site effects may indicate the importance of unmeasured management practises, the age of
the apple trees, the availability of elements and nutrients, pests and pathogens in the orchard and light
levels reaching the apples [eg. 30]. However, even considering site effects, pollination services in
orchards evidently needs to be considered in the management since it affects both fruit set and several
quality aspects of apples. To increase or obtain high natural pollination services in orchards, high
species richness and abundance of wild bees, seems to be the key [43-45]. Higher species richness of
pollinators have generally higher functional trait diversity and can therefore complement each other
both spatially and temporally leading to increased pollination services [43, 45, 46]. Moreover, the
effectiveness of managed bees as crop pollinators is enhanced by the presence of other interacting wild
pollinators [47].

In order to increase sustainability of apple production and to understand the full economic effect of
pollination services, we need a better understanding on the relationships between pollination services,
fruit set, marketable fruit quality and storable fruit quality. In our study, we found high pollination
levels to be beneficial not only for fruit set but also for higher DMC in fruit. The link between
pollination levels and storability were more complicated and need to consider the modifying effect of
fruit weight on the effect of pollination. However, lighter apples and lower K:Ca ratio in
supplementary hand-pollinated fruits indicate that these fruits have longer storability than less
pollinated fruits. Further research is needed to evaluate how general our results are and how they
interact with other management decisions such as selection of cultivars, thinning-regimes, and
pollinator management.

**Data accessibility.** All data and code associated with this manuscript are available via the Dryad
Digital Repository <https://doi:10.5061/dryad.nk9871p> [48]. (temporary review link:
<https://datadryad.org/review?doi=doi:10.5061/dryad.nk9871p>)

**Animal ethics.** No animal experiments are included in this study

**Permission to carry out fieldwork.** We had permission from the orchard owner to conduct field work
on his land.

**Funding.** This study was supported by grants from Ekhagastiftelsen (to US), the 2013-2014
BiodivERsA/FACCE-JPI joint call for research proposals (agreement # BiodivERsA-FACCE2014-
74), with the national funder FORMAS under grant no 226-2014-1784 (to PH), and from Lund
University (to HS). The research was part of the research environment BECC (Biodiversity and
Ecosystem Services in a Changing Climate).

**Competing interests.** We have no competing interests.

**Authors' contributions:** US developed the method and project idea. US collected and analysed the
data. US, PH, HS interpreted the results and conceptualized the frame for the manuscript. US wrote
the first drafts of the manuscript. All authors contributed substantially to revisions and gave final
approval for publication.

**Acknowledgements.** We are very grateful to Kiviks Musteri AB for their cooperation and the hard
work by the field assistants Christine Sandberg and Lucy Seeger.

**References**

- Klein, A. M., Vaissiere, B. E., Cane, J. H., Steffan-Dewenter, I., Cunningham, S. A., Kremen, C.,
Tschardtke, T. 2007 Importance of pollinators in changing landscapes for world crops. *Proceedings of*
*the Royal Society B-Biological Sciences*. **274**, 303-313. (doi:10.1098/rspb.2006.3721)
Eilers, E. J., Kremen, C., Greenleaf, S. S., Garber, A. K., Klein, A. M. 2011 Contribution of
pollinator-mediated crops to nutrients in the human food supply. *PLoS One*. **6**, e21363.
(doi:10.1371/journal.pone.0021363)
Chaplin-Kramer, R., Dombeck, E., Gerber, J., Knuth, K. A., Mueller, N. D., Mueller, M., Ziv, G.,
Klein, A. M. 2014 Global malnutrition overlaps with pollinator-dependent micronutrient production.
*Proc. R. Soc. B*. **281**, 20141799. (doi:10.1098/rspb.2014.1799)
Klatt, B. K., Holzschuh, A., Westphal, C., Clough, Y., Smit, I., Pawelzik, E., Tschardtke, T. 2014
Bee pollination improves crop quality, shelf life and commercial value. *Proc. R. Soc. B*. **281**,
20132440. (doi:10.1098/rspb.2013.2440)
Bommarco, R., Marini, L., Vaissiere, B. E. 2012 Insect pollination enhances seed yield, quality, and
market value in oilseed rape. *Oecologia*. **169**, 1025-1032. (doi:10.1007/s00442-012-2271-6)
Brittain, C., Kremen, C., Garber, A., Klein, A. M. 2014 Pollination and plant resources change the
nutritional quality of almonds for human health. *PLoS One*. **9**, (doi:10.1371/journal.pone.0090082)
Garratt, M. P. D., Breeze, T. D., Jenner, N., Polce, C., Biesmeijer, J. C., Potts, S. G. 2014 Avoiding
a bad apple: Insect pollination enhances fruit quality and economic value. *Agric. Ecosyst. Environ.*
**184**, 34-40. (doi:10.1016/j.agee.2013.10.032)
Matsumoto, S., Soejima, J., Maejima, T. 2012 Influence of repeated pollination on seed number and
fruit shape of 'Fuji' apples. *Sci. Hortic. (Amst.)*. **137**, 131-137. (doi:10.1016/j.scienta.2012.01.033)

Wietzke, A., Westphal, C., Gras, P., Kraft, M., Pfohl, K., Karlovsky, P., Pawelzik, E., Tschamtko,
479 T., Smit, I. 2018 Insect pollination as a key factor for strawberry physiology and marketable fruit
quality. *Agric. Ecosyst. Environ.* **258**, 197-204. (10.1016/j.agee.2018.01.036)
McGregor, S. E. 1976 *Insect pollination of cultivated crop plants*. Agricultural Research Service,
US dep. of agriculture.
Ramírez, F., Davenport, T. L. 2013 Apple pollination: A review. *Sci. Hortic. (Amst.)*. **162**, 188-
203. (doi:10.1016/j.scienta.2013.08.007)
Leonhardt, S. D., Gallai, N., Garibaldi, L. A., Kuhlmann, M., Klein, A. M. 2013 Economic gain,
stability of pollination and bee diversity decrease from southern to northern Europe. *Basic Appl. Ecol.*
**14**, 461-471. (doi:10.1016/j.baae.2013.06.003)
Stern, R. A., Eisikowitch, D., Dag, A. 2001 Sequential introduction of honeybee colonies and
doubling their density increases cross-pollination, fruit-set and yield in 'Red Delicious' apple. *J.*
*Hortic. Sci. Biotechnol.* **76**, 17-23.
Ladurner, E., Recla, L., Wolf, M., Zelger, R., Burgio, G. 2004 *Osmia cornuta* (Hymenoptera
Megachilidae) densities required for apple pollination: a cage study. *J. Apic. Res.* **43**, 118-122.
Harker, F. R., Kupferman, E. M., Marin, A. B., Gunson, F. A., Triggs, C. M. 2008 Eating quality
standards for apples based on consumer preferences. *Postharvest Biol. Tec.* **50**, 70-78.
(doi:10.1016/j.postharvbio.2008.03.020)
Palmer, J. W., Harker, F. R., Tustin, D. S., Johnston, J. 2010 Fruit dry matter concentration: a new
quality metric for apples. *J. Sci. Food Agric.* **90**, 2586-2594. (doi:10.1002/jsfa.4125)
Harker, F. R., Carr, B. T., Lenjo, M., MacRae, E. A., Wismer, W. V., Marsh, K. B., Williams, M.,
White, A., Lund, C. M., Walker, S. B., *et al.* 2009 Consumer liking for kiwifruit flavour: A meta-

analysis of five studies on fruit quality. *Food Qual. Prefer.* **20**, 30-41.
(10.1016/j.foodqual.2008.07.001)
Saei, A., Tustin, D. S., Zamani, Z., Talaie, A., Hall, A. J. 2011 Cropping effects on the loss of
apple fruit firmness during storage: The relationship between texture retention and fruit dry matter
concentration. *Sci. Hortic. (Amst.)*. **130**, 256-265. (doi:10.1016/j.scienta.2011.07.008)
Musacchi, S., Serra, S. 2018 Apple fruit quality: Overview on pre-harvest factors. *Sci. Hortic.*
(*Amst.*). **234**, 409-430. (10.1016/j.scienta.2017.12.057)
Dris, R., Niskanen, R., Fallahi, E. 1998 Nitrogen and calcium nutrition and fruit quality of
commercial apple cultivars grown in Finland. *J. Plant Nutr.* **21**, 2389-2402.
(doi:10.1080/01904169809365572)
Moggia, C., Yuri, J. A., Pereira, M. Year Mineral content of different apple cultivars in relation to
fruit quality during storage. 2006: International Society for Horticultural Science (ISHS), Leuven,
Belgium; 2006. p. 265-272.
Fallahi, E., Conway, W. S., Hickey, K. D., Sams, C. E. 1997 The role of calcium and nitrogen in
postharvest quality and disease resistance of apples. *Hortscience*. **32**, 831-835.
Ferguson, I. B., Watkins, C. B. 1992 Crop load affects mineral concentrations and incidence of
bitter pit in cox orange pippin apple fruit. *J. Am. Soc. Hortic. Sci.* **117**, 373-376.
Faust, M., Shear, C. B. 1968 Corking disorders of apples - a physiological and biochemical review.
*Bot. Rev.* **34**, 441-469. (doi:10.1007/bf02859134)
Casero, T., Benavides, A. L., Recasens, I. 2010 Interrelation between fruit mineral content and pre-
harvest calcium treatments on 'Golden smoothie' apple quality. *J. Plant Nutr.* **33**, 27-37.
(doi:10.1080/01904160903391057)

Conway, W. S., Sams, C. E., Hickey, K. D. Year Pre- and postharvest calcium treatment of apple
fruit and its effect on quality. 2002: International Society for Horticultural Science (ISHS), Leuven,
Belgium; 2002. p. 413-419.
Glenn, G. M., Poovaiah, B. W. 1990 Calcium-mediated postharvest changes in texture and cell-
wall structure and composition in golden delicious apples. *J. Am. Soc. Hortic. Sci.* **115**, 962-968.
Porcel, M., Andersson, G. K. S., Palsson, J., Tasin, M. 2018 Organic management in apple
orchards: Higher impacts on biological control than on pollination. *J. Appl. Ecol.* **55**, 2779-2789.
(10.1111/1365-2664.13247)
Bramlage, W. J., Weis, S. A., Greene, D. W. 1990 Observations on the relationships among seed
number, fruit calcium, and senescent breakdown in apples. *Hortscience.* **25**, 351-353.
Volz, R. K., Tustin, D. S., Ferguson, I. B. 1996 Pollination effects on fruit mineral composition,
seeds and cropping characteristics of 'Braeburn' apple trees. *Sci. Hortic. (Amst.)*. **66**, 169-180.
(doi:10.1016/s0304-4238(96)00934-x)
Buccheri, M., Di Vaio, C. 2004 Relationship among seed number, quality, and calcium content in
apple fruits. *J. Plant Nutr.* **27**, 1735-1746. (doi:10.1081/lpla-200026409)
Swedish board of agriculture. 2013 Official statistics of Sweden – Number of fruit trees 2012.
Report No.: JO 33 SM 1301.
Swedish board of agriculture. 2013 Den svenska äppelodlingen växer. På tal om jordbruk –
fördjupning om aktuella frågor.
Pinheiro, J., Bates, D., DebRoy, S., Sarkar, D., R Core Team. Nlme: linear and nonlinear mixed
effects models. R package version 3.1-122 ed 2015.

R Core Team. R: A language and environment for statistical computing. Vienna, Austria: R
Foundation for Statistical Computing 2017.
Hothorn, T., Bretz, F., Westfall, P. 2008 Simultaneous Inference in General Parametric Models.
*Biometrical J.* **50**, 346-363. (doi:10.1002/bimj.200810425)
Bates, D., Maechler, M., Bolker, B., Walker, S. *_lme4*: Linear mixed-effects models using Eigen
and S4_. R package version 1.1-9 ed 2015.
Harrison, X. A. 2014 Using observation-level random effects to model overdispersion in count data
in ecology and evolution. *PeerJ.* **2**, e616. (doi:10.7717/peerj.616)
Therneau, T. A Package for Survival Analysis in S. version 2.38 ed 2015.
Volz, R. K., Ferguson, I. B., Bowen, J. H., Watkins, C. B. 1993 Crop load effects on fruit mineral-
nutrition, maturity, fruiting and tree growth of cox orange pippin apple. *J. Hortic. Sci.* **68**, 127-137.
Zuur, A. F., Ieno, E. N., Walker, N. 2009 *Mixed effects models and extensions in ecology with R.*
Springer-Verlag New York.
Shear, C. B. Year Interaction of nutrition and environment on mineral composition of fruits. 1980:
International Society for Horticultural Science (ISHS), Leuven, Belgium; 1980. p. 41-50.
Blitzer, E. J., Gibbs, J., Park, M. G., Danforth, B. N. 2016 Pollination services for apple are
dependent on diverse wild bee communities. *Agric. Ecosyst. Environ.* **221**, 1-7.
(doi:10.1016/j.agee.2016.01.004)
Russo, L., Park, M. G., Blitzer, E. J., Danforth, B. N. 2017 Flower handling behavior and
abundance determine the relative contribution of pollinators to seed set in apple orchards. *Agric.*
*Ecosyst. Environ.* **246**, 102-108. (doi:10.1016/j.agee.2017.05.033)

Martins, K. T., Gonzalez, A., Lechowicz, M. J. 2015 Pollination services are mediated by bee
functional diversity and landscape context. *Agric. Ecosyst. Environ.* **200**, 12-20.
(doi:10.1016/j.agee.2014.10.018)
Hoehn, P., Tschardtke, T., Tylianakis, J. M., Steffan-Dewenter, I. 2008 Functional group diversity
of bee pollinators increases crop yield. *Proc. R. Soc. B.* **275**, 2283-2291.
(doi:10.1098/rspb.2008.0405)
Brittain, C., Williams, N., Kremen, C., Klein, A.-M. 2013 Synergistic effects of non-*Apis* bees and
honey bees for pollination services. *Proceedings of the Royal Society B-Biological Sciences.* **280**,
20122767. (doi:10.1098/rspb.2012.2767)
Samnegård, U., Hambäck, P. A., Smith, H. G. 2019 Data from: Impact of pollination services to
apples goes beyond increased fruit set. *Dryad Digital Repository.* (doi:10.5061/dryad.nk9871p)

**Table 1.** The effect of the interaction between pollination treatment and the apple weight at harvest on
the element concentrations, analysed with linear mixed-effect models (lme, nlme-package in R) with
the random effect treeID (n = 190). When the interactions were non-significant, the effect of treatment
and apple weight is presented separately. P-values were obtained using log-likelihood tests. The
predictor variables total buds per tree, final fruit set per branch, cover colour of the apples and site
were also included in the lme and the p-values for these variables, obtained using log-likelihood tests,
are reported in the supplementary material (ESM table S3).

**Table 2.** The effect of element concentration on storage duration analysed with Cox proportional
hazards regression models (coxph-function, survival-package in R), where β is the regression
coefficient where a positive sign represent greater risk of postharvest disorders. HR and 95% CI is the
hazard ratio with its 95% confidence interval showing the effect size of the covariates. P-values for
elements and weight are obtained with Wald test, where z gives the Wald statistic. Bold p-values
represent significant relationships ($p < 0.05$). The last column shows if site has been included as a co-
590 variable, a stratification-variable or has been dropped from the model. 145 apples were included in the
591 models.

**Figure 1.** The effect of the treatments “pollinator exclusion”, “control” and “supplementary hand-
pollination” on A) the initial fruit set (number of initial fruits divided by number of flowers per
branch) and B) final fruit set (number of ripe fruits divided by number of flowers per branch), using
predicted values from the GLS-models. Bars represent model-estimated standard errors. The initial
fruit set was analysed with a generalized least square model (GLS) to allow for different variances
between pollination treatments. Both the model for the initial and final fruit set included site as an
additional predictor variable. In the supplementary hand-pollination treatment the fruit set was
calculated for the whole branch but only 5-10 flowers on the branch was supplementary hand-

pollinated. Estimated means per treatment and the standard errors in B) are back-transformed from
squared rooted values.

**Figure 2.** Dry matter content (%) in relation to (A) the number of days the fruits were in storage
before suffering from postharvest disorders (n = 37), and (B) healthy fruits from the categories initial
(directly after harvest) and final measured fruits (after 161-162 days in storage) (n = 153). Apples that
had started to shrivel were considered as healthy first class fruit if no other damages were detected on
the skin. Supplemented pollinated fruits are represented by dark gray, controls by gray and pollinator
excluded by white. In A) dots are jittered to make dots more visible.

**Figure 3.** Piecewise structural equation model exploring the relationship among pollination treatment,
fruit weight, fruit set, fresh weight K:Ca ratio, site and storage time. A) Initial SEM showing all tested
paths, and B) is the final analysed SEM after variable transformations, addition of a missing path, and
deletion of insignificant relationships. Solid arrows represent significant unidirectional relationships (p
<0.05), where black arrows represent a positive relationship, red arrows a negative relationship, grey
arrow an interaction term and the dashed lines the variables included in the interaction term. The
numbers in the boxes on top of the arrows represent the standardised path coefficients (for continuous
variables), the numbers in the box on top of the interaction term represent the slope estimates for the
different pollination treatments.

**Table 1.**

Response variable	Treatment * apple weight (df = 2)	Slope estimates \pm Std. Error			Treatment (df = 2)	Apple weight (df = 1)
		pollinator exclusion	control	supplementary hand-pollination		
Log(K_Ca)	p = 0.34				p = 0.057 poll ex: 3.5 ± 0.10 con: 3.5 ± 0.09 supp: 3.3 ± 0.12	p = 0.069
Ca*	p = 0.20				p = 0.10	p = 0.002
K	p = 0.025	-0.86 ± 0.42	-1.06 ± 0.36	0.69 ± 0.58		
Sqrt(Zn)	p = 0.012	-0.000054 ± 0.00029	-0.0011 ± 0.00025	-0.0011 ± 0.00042		
Sqrt(P)	p = 0.062				p = 0.16	p = 0.17
Log(Fe)	p = 0.30				p = 0.14	p = 0.99
Mg	p < 0.001	-0.071 ± 0.021	-0.10 ± 0.018	0.052 ± 0.030		
Log(B)	p = 0.32				p = 0.24	p = 0.19

*two outliers were removed

**Table 2.**

Element	B ± SE	HR	95% CI	z-statistic	P-value	Site
K:Ca						strata
Log(K:Ca) * weight	-0.012 ± 0.006	0.99	0.98 - 0.99	-1.99	0.046	
Log(K:Ca)	2.740 ± 1.092	15.48	1.82 - 131.74	2.51	0.012	
Weight	0.048 ± 0.023	1.05	1.00 - 1.10	2.29	0.035	
Ca*						strata
Ca	-0.019 ± 0.013	0.98	0.96 - 1.01	0.15	0.147	
K						co-variable
K	0.0022 ± 0.00089	1.002	1.0004 - 1.00393	2.43	0.015	
Weight	0.0078 ± 0.0032	1.01	1.00 - 1.01	2.25	0.015	
Zn						co-variable
Log(Zn)	1.83 ± 0.45	6.25	2.58 - 15.14	4.06	<0.001	
Weight	0.011 ± 0.0034	1.01	1.00 - 1.02	3.2	0.001	
P						co-variable
P	0.015 ± 0.0084	1.02	0.99 - 1.03	1.79	0.073	
Weight	0.0060 ± 0.0031	1.01	1.00 - 1.01	1.90	0.058	
Fe						strata
Fe * weight	0.0045 ± 0.0080	1.00	0.99 - 1.02	0.57	0.569	
Fe	-0.20 ± 1.33	0.82	0.060 - 11.06	-0.15	0.878	
Weight	0.00036 ± 0.0093	1.00	0.98 - 1.02	-0.038	0.969	
Mg						strata
Mg	0.013 ± 0.016	1.01	0.98 - 1.00	0.79	0.243	
Weight	0.0063 ± 0.0035	1.07	1.0 - 1.01	1.80	0.072	
B						strata
B	-0.13 ± 0.11	0.86	0.71 - 1.09	-1.21	0.228	
Weight	0.0056 ± 0.0033	1.01	1.00 - 1.01	1.71	0.087	

*two outliers were removed

**Figure 1.**

**Figure 2.**

**Figure 3.**

634

<https://mc.manuscriptcentral.com/rsos>

Figure 1

Figure 2

Figure 3

Appendix B

Ulrika Samnegard,
Armidale, NSW, Australia
2019-08-13

Response to Editor and Referees

We are glad that the reviewers found our study interesting and well written. We have now revised the ms. RSOS-190326 according to the very valuable and relevant comments from the editor and the two reviewers.

The title of the paper has been changed to “Pollination treatment affects fruit set and modifies marketable and storable fruit quality of commercial apples” in line with but slightly different to the suggestion from the editor, because conveys relevant information to prospective readers. However, we are prepared to use the title suggested if the editor so choses.

Reviewer one provided additional review comments in a pdf-file, with comments focusing on the incorrect use of the terms “pollination” and “pollination levels”. We fully agree that the terminology should be changed and we have addressed all suggestions by generally changing the term “pollination level” to “pollination treatment”, and related pollination treatment to probability of successful cross-pollination with compatible pollen. We have also changed the terminology and the interpretation of the results in the discussion accordingly. To further clarify the treatments used we have changed the name of the control treatment to “natural pollination”.

Reviewer two suggested a slight change of the data used for evaluating fruit set levels (to only use data on the actually hand-pollinated flowers instead of data for the entire branch). We agree on this, and consequently there has been a small change in the results, where we no longer find a difference in final fruit set between the naturally and hand-pollinated treatments. This changed result is discussed, but do not impact the main results of the paper. All figures have been updated accordingly and the earlier analysis and figure is provided in the supplementary material.

Please find our responses to the referees’ specific comments below, our responses are in bold below each comment.

Ulrika Samnegård

In addition to addressing all of the reviewers' and editor's comments please also ensure that your revised manuscript contains the following sections as appropriate before the reference list:...

We have ensured that the manuscript contains the section with ethics statement, data accessibility, competing interests, author's contributions, Acknowledgement and funding statement.

Subject Editor Comments to Authors:

Thanks for your submission. The reviewers found it very interesting, but also raised some issues that should be addressed, including the use of terminology and the study methods. Please address these specifically in your revisions.

Also, could I suggest altering the title somewhat? I found it difficult to interpret, and for a general biological journal like RSOS readers should not have to struggle. Maybe something like "Pollination increases more than fruit set rates in commercial apples." You could even add a colon and in a subtitle list a few of the other outcome variables that are affected. Thanks for considering, and best wishes with your revisions.

Thank you for the comments and guidance on what to focus the revision on. We have now changed the terminology throughout the manuscript to instead of using the term "pollination level" we use "pollination treatment" and we discuss how this relates to successful cross-pollination and fertilization. We have also re-analysed the initial and final fruit set data according to the suggestions from reviewer two. We also changed the title in line with the suggestion by the editor to "Pollination treatment affects fruit set and modifies marketable and storable fruit quality of commercial apples".

Reviewers' Comments to Author:

Reviewer: 1

Review of "Impact of pollination services to apples goes beyond increased fruit set"

Greetings editors and authors:

It always continues to amaze me that in such a well-studied crop like apple, there are still so many things we don't know about the reproductive biology of the plant, and the ultimate impacts on fruit quality. In this study, the authors investigate the impact of different pollination treatments (bagged flowers = no pollinators access; open pollination; and hand pollination with hand collected compatible pollen), on several metrics of fruit quantity (i.e., fruit set) and quality (i.e., fruit weight, element content, dry matter content (DMC) and K:Ca ratio). For some of these variable seem to have little bearing on fresh fruit marketing, they do indeed play a big role in the post-harvest physiology of the fruit, and are critical for extending the storability of the fruit which is important for having apples available to markets throughout the year. Thus, I think this manuscript will ultimately be of interest to many. I also think that it is very well-written, and my main concerns for its publication are only with the use of terms.

My biggest concern about this manuscript is that the authors use terminology associated with the word "pollination" incorrectly. To be as simple as possible, pollination simply refers to the delivery of pollen from anthers to stigmas, nothing more, nothing less. However, this means that one can have an apple flower that is adequately pollinated, but which produces no fruit, a misshaped fruit, or an excellent quality fruit – thus fruit quality, including the variables studied here may be more related to post-pollination events (i.e., fertilization) resulting from pollination with compatible pollen. Remember pollination can also occur with non-compatible pollen, and even self-pollination is still pollination, even though it may not produce fruit.

Thank you for providing this very important comment on our incorrect use of the term "pollination". We fully agree with you and have change the terms and wording so we do no longer discuss pollination level. Instead we focus on pollination treatment and how it may be related to cross-pollination with compatible pollen.

Throughout the manuscript, the authors use the term "pollination levels" which is not correct; this term should be used for when one assessed the actual number of pollen grains on the stigmas, which was not done here. This is not trivial; in one of my previous studies (Sheffield et al. 2005) I demonstrated that for many apple cultivars, the notion that pollen deposition needs to be heavy, and evenly distributed on the five stigmas of a flower is not necessarily true. Pollination can occur on only one of the five stigmatic surfaces to produce a good quality fruit due to an internal connection in the apple flower. As such, pollination does not have to be full, in fact, in theory five to ten compatible pollen grains on one stigmatic surface could produce a well-shaped fruit with a full complement of seeds. However, most flowers will receive 100s of pollen grains, though this is influenced by weather, and stage of flowering (see Sheffield 2014). The number of seeds does not have to be maximized, so the relationship between seed number and fruit size and shape is not always easy to interpret. In another work I (Sheffield 2014) showed that seed distribution, and not necessarily number was most important for fruit quality. I am happy to share these works if the authors wish to see them.

We agree, we did not count actual pollen grains on stigmas and should therefore not use the term "pollination level". We have changed accordingly.

Thus, as an alternative to "pollination levels" I suggest that the authors use "pollination treatments" consistently throughout, the treatments being the control (bagged flowers), natural pollination, and hand pollination with compatible pollen. As such, you should also discuss that the "natural pollination" by bees was not necessarily the issue, but rather the pollen they were delivering itself. Pollinators don't care about the source of the pollen, so may be moving around compatible or non-compatible pollen; depending on the cultivar mix, distribution, and overlapping flowering phenology, the bees may or may not be moving around suitable pollen, but that does not mean they are not adequately pollinating the blossoms. Thus the authors should be stressing pollination treatments having the impact (due to pollen compatibility, likely being better in the hand pollination treatments), not pollination levels, as the latter was not investigated. As such, the pollinators are likely doing a great job, and it may be that other factors are involved.

We have changed "pollination levels" to "pollination treatments" and also the name of the control treatments to make it clearer for the reader. Now the different treatments are called "pollinator exclusion", "natural pollination" and "supplementary hand-pollination". We expanded a paragraph in the discussion to also discuss how to increase successful cross-pollination:

Hence, even considering site effects, increasing cross-pollination in orchards evidently needs to be considered in the management since it affected several quality aspects of apples. To increase successful cross-pollination from natural pollination, orchards need to have ample compatible pollen sources in configurations that facilitate pollinator movements between compatible cultivars [45]

In this context, I think the authors should re-think their results based on pollen compatibility, not pollination levels, with respect to the metrics used to assess fruit quality. Remember that not all pollen is equal; studies in Canada by Kron et al. (cited in papers below) determined that competition between pollen types on a stigma occurs internally, so the desired impact on fruit quality via higher DMC and lower K:Ca ratio may be the result of the types of donor pollen(s), not the presence of the pollen itself (so, not necessarily a pollination issue). I think then the differences observed are not pollinator related, but linked to orchard design, cultivar mix and orchard layout, and pollinator movement patterns (they move down rows). Thus (and for example), from line 374 "...indicates that supplementary hand-pollinated apples could have a generally better storability" is only true in the context of the pollen, not pollination per say. You cannot state that pollination is better at all, as you did not look at that. All you can confirm is that when ample compatible pollen is delivered, then storability is better. This should not be seeming to suggest that bees are not performing.

Thank you for the suggested reference; we refer to it and have also changed the wording in the results and discussion. As mentioned above, we have also changed to discussing cross-pollination instead of pollination levels in orchards.

Otherwise, I enjoyed this research, and hope the authors will find these comments (which are also on the attached manuscript) useful.

Thank you!

Reviewer: 2

Comments to the Author(s)

This manuscript explores the importance of pollination for fruit production in commercial apple orchard. It extends previous research by asking how the level of pollination influencing the dry matter weight and mineral composition of the apples, rather than simply focusing on fruit and seed set. The

results show that additional hand pollination increase the dry matter content and reduced the K/Ca ratio of fruits, components which are positively associated with greater storability and consumer preference.

Structural equation models were used to demonstrate that pollination influences the storability of apples through indirect mechanisms involving the mineral uptake into the fruits, a process that varies with pollination treatment and weight.

The study highlights that there is more to learn about the benefits that effective pollination can provide for commercial fruit growers.

The methods seem robust and have been analysed with appropriate statistical models. In places the manuscript would benefit from being made more concise, particularly in the discussion which has a tendency to repeat the results. I have a few queries about the methodology and minor suggestions which are listed below.

Abstract

Line 19 change increase to "can increase" or "increases". Same with influence in the following line.

Changed accordingly

Methods

Line 138. You only hand pollinated 5 or so flowers per branch. It is clear that you only collected and weighed the fruits from the hand-pollinated flowers, but when calculating fruit set it sounds as if you counted all flowers and fruits on the branch. This seems inappropriate as the majority of flowers on the branch were not hand-pollinated so are equivalent to the control. Please could you clarify this.

This is correct – we have now re-analysed the fruit set data to only include the hand-pollinated fruits. Results and figures are updated accordingly and the branch-level analysis is found in the supplementary material (figure s1). The results changed slightly: we do no longer find a difference in final fruit set between the naturally and hand-pollinated treatment, which is discussed in the Discussion:

“Even though pollination treatment affected both quantity and several quality aspects of apples, other site factors also had a strong impact, as shown by the significant site effect in several of the analyses (e.g. figure 3, ESM table S3). Site effects may indicate the importance of unmeasured management practises like thinning and fertilization regimes, and site variation in the age of the apple trees, the availability of elements and nutrients in the soil, pests and pathogens and other animal interactions in the orchard, and light levels reaching the apples [eg. 30, 43, 44]. These unmeasured factors may have concealed differences in final fruit set levels between natural and supplementary hand-pollinated fruits, while the effect of the pollinator treatments persisted for the quality variables.”

Line 184 – Why did you exclude apples that did not meet the marketing standards? In line with your working hypothesis you might expect under-pollinated fruits to fall below the standards more frequently than those in the hand-pollination treatments so you seem to be excluding useful data without a clear explanation.

Since our study was focusing on the storability of apples and the fresh market, we found it appropriate to only work with fruits that actually would have been considered to be sold in the fresh market. Fruits that doesn't meet the standards are disregarded to juice or cider production and would not have been stored. Even though it would have been very interesting to evaluate the proportion of fruits meeting the EU standards from the different treatments

that would have required a different sampling design. We only picked the finest apples per branch, those we thought met the standards and hence were suitable for storing. Hence the sampling was biased towards the finest fruits and not all fruits were assessed for their quality. However, our fruit set measurements include all fruits on the branches.

Line 213 – Please could you clarify what “possible dilution effect” and “initial investment by tree” are.

Since “possible dilution effect” and “initial investment by tree” are only speculations on why the different variables could have an effect on the response variables we delete those phrases from the text to not confuse the reader.

Line 247 I think hazard rate should be replaced by hazard ratio?

Clarified to: “A ratio of the hazard rates (referred to as hazard ratio) >1 indicates that a variable has a negative influence on storage duration”

Results

Line 280. Again, am I right in thinking only 5 flowers per branch received additional hand-pollination, but that fruit set was calculated for the entire branch? This does not seem appropriate and be at least be justified and discussed.

Correct and changed, see replay to comments above

Line 312 This [paragraph would benefit from rewording. Currently the explanation of the post hoc test results is unclear and a bit confusing.

The paragraph has been changed and is hopefully clearer now.

332. I think it is useful to use Structural Equation Model rather than SEM in the sub-title as other authors some times use this acronym for standard error of the mean.

Changed accordingly

Discussion

Line 359 Influence the risk in which direction?

Changed and clarified: “Our studies show that also higher concentration of Zn, and not only K and higher K:Ca ratio, may influence the risk of postharvest disorders during storage.”

Line 372. I suggest that you avoid referencing Figures and tables in the discussion. They are best kept to the results section.

Changed, references to tables and figures are removed

Line 387 Found to predict it better than what?

Changed and clarified: *“Since fruits are harvested before starch solubilisation is completed, the soluble solids at harvest do not represent the soluble solids after storage well [16]. Measurements of DMC at harvest have instead been found to better predict the total soluble solids in fruits after storage, probably since DMC also consider starch levels [16]”*

The end of this paragraph could be made more concise. At the moment it is challenging to follow.

The last sentence is rewritten: *“On the other hand, the retained higher DMC throughout the storage time in supplemented hand-pollinated fruits, indicated a persistently higher quality of ensured cross-pollinated fruits compared to the other treatments.”*

Line 395 found should be changed to find

Changed accordingly

Line 399 Parts of your discussion seem to be repeating the results section would benefit from being condensed, with the focus shifted towards interpreting rather than repeating the results

The discussion has partly been re-written and references to tables and figures are removed. However due to the complexity of the analyses and results, we feel that we need to go through the results thoroughly to ease the understanding of the reader.

Line 406 Change found to find.

Changed accordingly

Figures and Tables are all clear and well presented, but it is not necessary to provide so much detail about the statistical methods in the figure legends as this information is better suited in the methods.

Thank you! Some details about the statistical methods are removed from the legends.